# ALIGNER, DIAGNOSE THYSELF: A META-LEARNING PARADIGM FOR FUSING INTRINSIC FEEDBACK IN PREFERENCE ALIGNMENT

**Mengyang Li[1]    Pinlong Zhao[2]    Zhong Zhang[1],***

[1]Tianjin Key Laboratory of Wireless Mobile Communications and Power Transmission,
Tianjin Normal University, Tianjin, China, 300387
[2]School of Cyberspace, Hangzhou Dianzi University, Hangzhou, China, 310018
`limengyang@tjnu.edu.cn, pinlongzhao@hdu.edu.cn`
`zhong.zhang8848@gmail.com`

## ABSTRACT

The alignment of Large Language Models (LLMs) with human preferences is critically undermined by noisy labels in training datasets. Existing robust methods often prove insufficient, as they rely on single, narrow heuristics such as perplexity or loss, failing to address the diverse nature of real-world noise. We challenge this limited-scope approach by introducing a new paradigm where models learn to diagnose thyself, systematically fusing multiple streams of intrinsic feedback for a holistic reliability assessment of each preference pair. We instantiate this paradigm through a meta-learning methodology that learns to adaptively reweight samples based on a rich diagnostic vector. This vector captures three complementary perspectives: preference consistency, learning difficulty, and generation confidence. Extensive experiments demonstrate that our approach significantly outperforms state-of-the-art methods across various noise conditions. Crucially, our work provides the first quantitative analysis of these intrinsic diagnostics, revealing that their fusion is essential for overcoming the blind spots inherent in any single heuristic. This diagnostic-driven paradigm offers a principled path towards developing more robust and trustworthy LLMs.

## 1 INTRODUCTION

Large Language Models (LLMs) have demonstrated remarkable capabilities across a wide range of tasks (Brown et al., 2020; Touvron et al., 2023). A crucial step in realizing their full potential lies in aligning these models with human preferences, ensuring they are helpful, harmless, and honest (Cao et al., 2021; Bai et al., 2022). This alignment process often relies on preference datasets, where humans or AI systems indicate preferred responses among candidate pairs (Christiano et al., 2017; Stiennon et al., 2020; Rafailov et al., 2023). However, these datasets are often plagued by *noisy preferences* (NPs), where the recorded preference label is incorrect due to annotator disagreement, subjective biases, or errors in AI-based labeling (Gao et al., 2024; Zheng et al., 2023; Baumgärtner et al., 2024). NPs can severely degrade alignment quality, leading to poor model performance and the reinforcement of undesirable behaviors (Gao et al., 2024; Rafailov et al., 2023).

To mitigate the impact of NPs, existing robust alignment methods can be broadly categorized into two groups. The first group employs *coarse-grained adjustments*, such as modifying the loss function with global noise estimates (Rafailov et al., 2023; Chowdhury et al., 2024). While offering some robustness, these methods lack the precision to handle instance-specific noise, treating all samples equally regardless of their individual characteristics. The second group leverages *single-heuristic criteria* to identify and correct or down-weight potentially noisy samples. A prominent example is Kong et al. (2024), who utilize the perplexity difference (PPLDiff) between preferred and dispreferred responses as a signal for label inconsistency. While these methods offer instance-level granularity, they rely on a single, often myopic heuristic, neglecting the multifaceted nature

---

*Corresponding author.

of preference reliability. For instance, PPLDiff may be misleading when dealing with fluent but factually incorrect responses, or when the model is inherently uncertain about the best response due to subjective or nuanced queries.

To address the limitations of existing approaches, we introduce a new paradigm for robust preference alignment: *Aligner, Diagnose Thyself.* Instead of relying on a single, potentially flawed signal, our paradigm empowers the alignment model to act as its own diagnostician, systematically fusing multiple streams of *intrinsic feedback* to assess the reliability of each preference pair. We argue that preference reliability is not a monolithic property, but rather a multifaceted construct that can be best understood by considering complementary perspectives derived from the model's internal state. Specifically, we identify three key perspectives that, when combined, provide a more holistic and reliable assessment of preference quality:

- **Preference Consistency:** Does the model's likelihood estimation align with the provided preference label? This is captured by the dynamic perplexity difference (PPLDiff) between the preferred and dispreferred responses, reflecting the model's intrinsic assessment of their fluency and plausibility (Kong et al., 2024).

- **Learning Difficulty:** How easily does the model assimilate the preference information? This is quantified by the training loss incurred by the preference pair, reflecting the degree to which the sample aligns with the model's current understanding of the task (Ren et al., 2018; Shu et al., 2019). High loss values may indicate noisy labels or challenging edge cases.

- **Generation Confidence:** How certain is the model in its generation process? This is estimated by the uncertainty associated with the model's token predictions, reflecting the model's internal confidence in its chosen responses. High uncertainty may suggest that the model is struggling to distinguish between plausible alternatives, potentially indicating subjective or ambiguous preferences.

These three perspectives, while individually informative, are inherently limited. PPLDiff can be misled by fluent misinformation, loss can be high for both noisy and genuinely difficult examples, and uncertainty can arise from both ambiguity and a lack of knowledge. Therefore, a robust alignment method must be able to intelligently fuse these diverse and sometimes conflicting signals to arrive at a more informed assessment of preference reliability.

We operationalize this paradigm with a novel meta-learning methodology that learns to adaptively reweight training samples based on a rich diagnostic vector. This vector captures the three aforementioned intrinsic feedback signals, allowing the model to dynamically adjust its learning process based on its own self-assessment. By training a meta-learner to interpret this diagnostic vector and assign appropriate weights, we enable the model to prioritize reliable preferences while down-weighting potentially noisy ones.

The contributions of this work are as follows:

- We introduce a new paradigm for robust preference alignment based on fusing multiple intrinsic model diagnostics, empowering models to diagnose thyself instead of relying on single, potentially flawed heuristics.

- We instantiate this paradigm with a novel meta-learning methodology that learns to weigh samples based on a diagnostic vector capturing preference consistency, learning difficulty, and generation confidence.

- We provide the first systematic analysis of the interplay and relative importance of these intrinsic diagnostics, revealing that their fusion is essential for overcoming the limitations inherent in any single heuristic.

- We conduct comprehensive experiments to demonstrate our method's superiority over state-of-the-art baselines under various noise conditions, including strong baselines utilizing perplexity differences.

## 2    RELATED WORK

Our work is situated at the intersection of three research areas: robust preference alignment for LLMs, learning with noisy labels, and meta-learning for robustness.

**LLM Alignment with Noisy Preferences.**    Aligning LLMs with human values via preference data is a cornerstone of modern AI safety (Ouyang et al., 2022; Bai et al., 2022), with methods like Reinforcement Learning from Human Feedback (RLHF) (Christiano et al., 2017; Stiennon et al., 2020) and Direct Preference Optimization (DPO) (Rafailov et al., 2023) being widely adopted. However, the susceptibility of these methods to noisy preferences is a well-documented challenge (Gao et al., 2024; Zheng et al., 2023). Initial efforts to address this focused on robust loss formulations. For instance, cDPO (Rafailov et al., 2023) and rDPO (Chowdhury et al., 2024) introduce confidence scaling or label smoothing based on a global noise estimate, applying a uniform correction across all samples. More recently, methods have shifted towards instance-level heuristics. A notable example is PerpCorrect (Kong et al., 2024), which uses the perplexity difference (PPLDiff) between responses as a direct signal to detect and flip potentially mislabeled preferences. While effective, these approaches remain confined to a single diagnostic perspective.

**Learning with Noisy Labels (NLL).**    The problem of learning from corrupted supervision is a long-standing challenge in machine learning (Frénay & Verleysen, 2013; Song et al., 2022). Sample reweighting is a prominent paradigm within NLL, where the core idea is to down-weight instances that are likely to be mislabeled (Liu & Tao, 2015; Jiang et al., 2018). Various strategies have been proposed to determine these weights, often based on heuristics like the training loss of a sample—the intuition being that noisy samples tend to have higher loss values (Han et al., 2018; Shu et al., 2019; Li et al., 2025). Our work adapts this established sample reweighting principle to the unique context of LLM preference alignment.

**Meta-Learning for Robustness.**    Meta-learning, or "learning to learn", has proven to be a powerful technique for designing adaptive training algorithms (Ren et al., 2018). In the context of robust learning, a particularly successful application has been to meta-learn a sample reweighting function (Shu et al., 2019; Jamal et al., 2020). These methods typically train a small neural network (a "meta-net") to map features like training loss to sample weights, with the meta-net's parameters being optimized based on performance on a small, clean meta-dataset. Our methodology builds directly upon this meta-learning foundation. Our key novelty lies in *what* we feed into the meta-learner.

## 3    METHODOLOGY

In this section, we formally introduce our paradigm for robust preference alignment. We begin by defining the problem setup and then detail the construction of our multi-perspective diagnostic vector. Finally, we present our meta-learning methodology for learning to fuse these diagnostics to achieve robust alignment. The overall architecture of our approach is illustrated in Figure 1.

### 3.1    PRELIMINARIES: DIRECT PREFERENCE OPTIMIZATION

We build upon the Direct Preference Optimization (DPO) framework (Rafailov et al., 2023). Let $\mathcal{D} = \{(x^{(i)}, y_w^{(i)}, y_l^{(i)})\}_{i=1}^{N}$ be a preference dataset, where $x^{(i)}$ is a prompt, $y_w^{(i)}$ is the preferred response, and $y_l^{(i)}$ is the dispreferred response. DPO aims to train an LLM policy $\pi_\theta$ to satisfy these preferences, starting from a reference policy $\pi_{\mathrm{ref}}$ (typically an SFT model). The DPO loss for a single preference pair is given by:

$$\mathcal{L}_{\mathrm{DPO}}(\pi_\theta, \pi_{\mathrm{ref}}) = -\log \sigma \left( \beta \log \frac{\pi_\theta(y_w|x)}{\pi_{\mathrm{ref}}(y_w|x)} - \beta \log \frac{\pi_\theta(y_l|x)}{\pi_{\mathrm{ref}}(y_l|x)} \right) \tag{1}$$

where $\beta$ is a hyperparameter that controls the deviation from the reference policy, and $\sigma(\cdot)$ is the logistic function. When the training dataset $\mathcal{D}$ contains noisy preferences (i.e., $(y_w, y_l)$ are swapped), directly minimizing this loss can lead the model to learn incorrect behaviors.

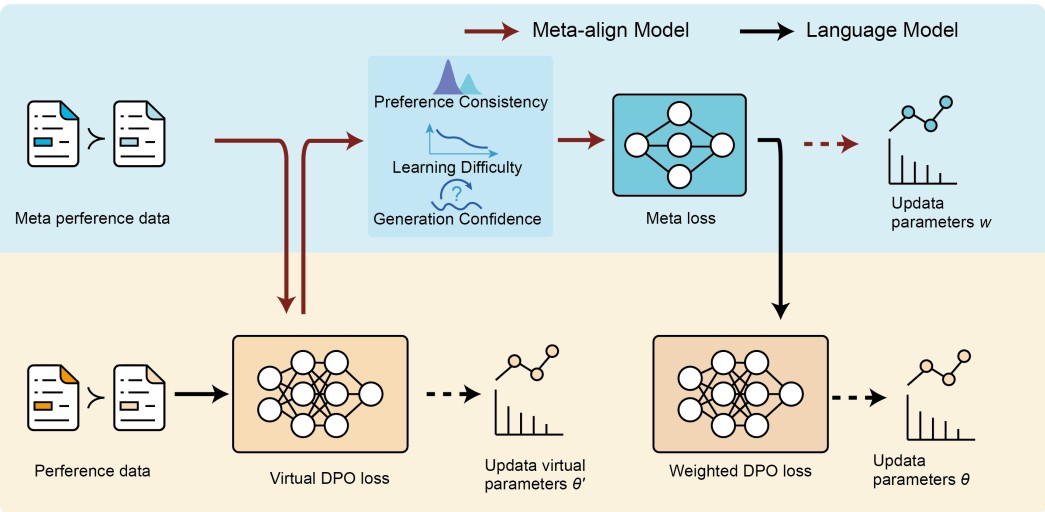

Figure 1: An overview of our diagnostic-driven meta-learning paradigm.

## 3.2 CONSTRUCTING THE INTRINSIC DIAGNOSTIC VECTOR

Our core premise is that a model's own internal state provides rich, multi-faceted feedback about the reliability of a given preference pair. We capture this feedback in a dynamic *intrinsic diagnostic vector*, $\mathbf{z} \in \mathbb{R}^d$, computed for each sample at every training step. This vector comprises three key components, each offering a complementary perspective on data quality.

**Preference Consistency ($z_{\mathbf{ppl}}$).** A well-aligned model should assign a higher likelihood (lower perplexity) to a genuinely preferred response. A deviation from this expectation is a strong indicator of a potential label-model conflict. We quantify this using the log-perplexity difference (PPLDiff), computed dynamically with the current policy $\pi_{\theta_t}$ at training step $t$:

$$z_{\mathrm{ppl}}^{(i)} = \log \mathrm{PPL}(\pi_{\theta_t}, [x^{(i)}; y_w^{(i)}]) - \log \mathrm{PPL}(\pi_{\theta_t}, [x^{(i)}; y_l^{(i)}]), \tag{2}$$

where $\mathrm{PPL}(\pi, s) = \exp(-\frac{1}{|s|} \sum_{k=1}^{|s|} \log \pi(s_k|s_{<k}))$. A positive $z_{\mathrm{ppl}}^{(i)}$ suggests that the labeled winning response is less likely under the current model than the losing one, flagging it as a potential NP.

**Learning Difficulty ($z_{\mathbf{loss}}$).** The magnitude of the training loss for a sample reflects how inconsistent it is with the model's current parameterization. Noisy samples often present conflicting gradients, resulting in higher loss values. We use the instance-wise DPO loss itself as a signal of learning difficulty:

$$z_{\mathrm{loss}}^{(i)} = \mathcal{L}_{\mathrm{DPO}}(\pi_{\theta_t}, \pi_{\mathrm{ref}}; (x^{(i)}, y_w^{(i)}, y_l^{(i)})). \tag{3}$$

This provides a direct measure of how surprising a given preference is to the model.

**Generation Confidence ($z_{\mathbf{uncert}}$).** Beyond likelihood, the model's confidence during the generation process offers another valuable signal. A model that is uncertain about its token predictions for a given response may be grappling with ambiguity or subjectivity in the prompt, making the corresponding preference label less reliable. We measure this confidence using the average token-level entropy of the generated responses. Specifically, for a response $y = (y_1, \ldots, y_m)$, the uncertainty is:

$$H(y|x; \pi_{\theta_t}) = -\frac{1}{m} \sum_{j=1}^{m} \sum_{v \in \mathcal{V}} \pi_{\theta_t}(v|x, y_{<j}) \log \pi_{\theta_t}(v|x, y_{<j}), \tag{4}$$

where $\mathcal{V}$ is the vocabulary. High entropy indicates low confidence. We use the uncertainty of the preferred response as our diagnostic signal, $z_{\mathrm{uncert}}^{(i)} = H(y_w^{(i)}|x^{(i)}; \pi_{\theta_t})$, as noisy preferences often correspond to less coherent or confident generations for the supposed winner.

---

**Algorithm 1** Meta-Learning for Fusing Intrinsic Diagnostics

---

**Input:** Noisy data $\mathcal{D}$, clean meta-data $\mathcal{D}_{\text{meta}}$, initial $\theta_0, W_0$, rates $\alpha_\theta, \alpha_W$, steps $T$.
**Output:** Aligned model parameters $\theta_T$.
 1: **for** $t = 0$ to $T - 1$ **do**
 2:      Sample mini-batches $\mathcal{B}_t \subset \mathcal{D}$ and $\mathcal{B}_{\text{meta}} \subset \mathcal{D}_{\text{meta}}$.
 3:      Compute diagnostic vectors $\{\mathbf{z}_t^{(j)}\}_{j \in \mathcal{B}_t}$ using $\theta_t$.                   ▷ Sec. 3.2
 4:      Compute weights $\{v_t^{(j)} = V(\mathbf{z}_t^{(j)}; W_t)\}_{j \in \mathcal{B}_t}$.
 5:      Compute virtual parameters $\theta_t'(W_t)$ via Eq. 6.
 6:      Compute meta-loss $\mathcal{L}_{\text{meta}}(W_t)$ on $\mathcal{B}_{\text{meta}}$ using $\theta_t'(W_t)$ via Eq. 7.
 7:      Update meta-learner: $W_{t+1} \leftarrow W_t - \alpha_W \nabla_{W_t} \mathcal{L}_{\text{meta}}(W_t)$.
 8:      Update main model: $\theta_{t+1} \leftarrow \theta_t - \alpha_\theta \nabla_{\theta_t} \mathcal{L}_{\text{weighted}}(\theta_t, W_{t+1})$.      ▷ Using new weights
 9: **end for**
10: **return** $\theta_T$.

---

The final diagnostic vector for sample $i$ at step $t$ is the concatenation of these normalized components: $\mathbf{z}_t^{(i)} = [\text{norm}(z_{\text{ppl}}^{(i)}), \text{norm}(z_{\text{loss}}^{(i)}), \text{norm}(z_{\text{uncert}}^{(i)})]$.

### 3.3 A Meta-Learning Formulation for Fusing Diagnostics

Given the diagnostic vector $\mathbf{z}$, our goal is to learn a function $V(\mathbf{z}; W)$ that maps these diagnostics to a non-negative sample weight, where $W$ are the parameters of the meta-learner. We employ a meta-learning strategy (Ren et al., 2018) where the quality of the weights produced by $V$ is evaluated based on the main model's performance on a small, clean meta-dataset, $\mathcal{D}_{\text{meta}}$. This bi-level optimization can be understood as learning an implicit, adaptive weighting scheme, with theoretical guarantees that minimizing the empirical meta-loss leads to good generalization on the true clean data distribution. We provide a detailed theoretical analysis in Appendix A.

The training proceeds in a bi-level optimization loop. At each step $t$, we sample a mini-batch $\mathcal{B}_t$ from the noisy training set $\mathcal{D}$ and a mini-batch $\mathcal{B}_{\text{meta}}$ from the clean meta-set $\mathcal{D}_{\text{meta}}$.

**Inner Loop: Virtual Update.** First, we compute the diagnostic vector $\mathbf{z}_t^{(j)}$ for each sample $j \in \mathcal{B}_t$ using the current policy $\pi_{\theta_t}$. The meta-learner $V(\cdot; W_t)$ then produces weights $v_t^{(j)} = V(\mathbf{z}_t^{(j)}; W_t)$. These weights modulate the DPO loss on the training batch:

$$\mathcal{L}_{\text{weighted}}(\theta_t, W_t) = \frac{1}{|\mathcal{B}_t|} \sum_{j \in \mathcal{B}_t} v_t^{(j)} \mathcal{L}_{\text{DPO}}(\pi_{\theta_t}, \pi_{\text{ref}}; j). \tag{5}$$

We then compute a hypothetical one-step gradient update for the main model, resulting in virtual parameters $\theta_t'(W_t)$:

$$\theta_t'(W_t) = \theta_t - \alpha_\theta \nabla_{\theta_t} \mathcal{L}_{\text{weighted}}(\theta_t, W_t), \tag{6}$$

where $\alpha_\theta$ is the learning rate for the main model.

**Outer Loop: Meta-Objective and Updates.** The quality of the weighting parameters $W_t$ is assessed by evaluating the performance of the virtual model $\pi_{\theta_t'(W_t)}$ on the clean meta-batch $\mathcal{B}_{\text{meta}}$. This yields the meta-loss:

$$\mathcal{L}_{\text{meta}}(W_t) = \frac{1}{|\mathcal{B}_{\text{meta}}|} \sum_{k \in \mathcal{B}_{\text{meta}}} \mathcal{L}_{\text{DPO}}(\pi_{\theta_t'(W_t)}, \pi_{\text{ref}}; k). \tag{7}$$

The meta-learner's parameters $W$ are then updated by descending the gradient of this meta-loss: $W_{t+1} = W_t - \alpha_W \nabla_{W_t} \mathcal{L}_{\text{meta}}(W_t)$. Finally, the main model's parameters $\theta_t$ are updated using the original training batch $\mathcal{B}_t$, but with weights computed from the *updated* meta-learner $V(\cdot; W_{t+1})$:

$$\theta_{t+1} = \theta_t - \alpha_\theta \nabla_{\theta_t} \mathcal{L}_{\text{weighted}}(\theta_t, W_{t+1}). \tag{8}$$

This process, summarized in Algorithm 1, allows the meta-learner to learn an effective, data-driven strategy for fusing the intrinsic diagnostics, guided by the objective of improving performance on clean, reliable data.

## 4 EXPERIMENTS

We conduct a comprehensive set of experiments to validate our proposed paradigm. Our evaluation is designed to answer three key questions: (1) Does our diagnostic fusion approach outperform state-of-the-art robust alignment baselines across various noise conditions? (2) What is the individual contribution of each intrinsic diagnostic, and is their fusion truly necessary? (3) How do the different diagnostics interact, and what is their relative importance in identifying noisy preferences?

### 4.1 EXPERIMENTAL SETUP

**Datasets and Noise Simulation.**    Our experiments are conducted on two widely-used public preference datasets: Golden HH (Bai et al., 2022; Ethayarajh et al., 2024), a helpfulness-focused subset of Anthropic-HH, and OASST1 (Köpf et al., 2023), a multi-turn conversational dataset. To evaluate robustness under controlled conditions, we simulate noisy preferences by randomly swapping the 'chosen' and 'rejected' labels for a fraction $\epsilon \in \{0.1, 0.2, 0.3, 0.4\}$ of the training samples, following standard protocols (Kong et al., 2024; Chowdhury et al., 2024). To validate scalability and real-world applicability beyond synthetic noise, we additionally conduct experiments on two large-scale datasets: HuggingFace H4 StackExchange Preferences (10.8M samples) containing community-voted technical Q&A with inherent annotation subjectivity, and GPT4All (0.8M samples) representing LLM-distilled preferences with generation artifacts. These datasets exhibit naturally occurring noise from human disagreement and distillation errors, providing a complementary testbed to synthetic label-flipping. For our method, a small, clean meta-dataset is held out from the original training set: $M = 100$ samples for Golden HH and OASST1, and $M = 200$ for the large-scale datasets. These sizes were chosen based on sensitivity analysis (see Appendix C), which shows that performance saturates in this range, making them practical and effective choices. Regarding the practical acquisition of such data, we explore scalable strategies including high-agreement filtering and expert curation in Appendix D.5. Further details on data splits are provided in Appendix B.1.

**Models and Implementation.**    We evaluate on a suite of open-source LLMs to demonstrate broad applicability, including Llama-2-7B (Touvron et al., 2023), Phi-2 (Javaheripi et al., 2023), and Llama-3-8B for large-scale and real-world evaluations (Tables 1–2). Our method and all DPO-based baselines are implemented using the TRL library (von Werra et al., 2020) for consistency. The meta-learner $V(\cdot; W)$ in our approach is a two-layer MLP. All hyperparameters and implementation details are detailed in Appendix B.2 to ensure full reproducibility.

**Baselines.**    We compare our approach against a strong and diverse set of baselines. These include Vanilla DPO (Rafailov et al., 2023); robust DPO variants such as cDPO (Rafailov et al., 2023), rDPO (Chowdhury et al., 2024), and the recent state-of-the-art DR-DPO (Azar et al., 2024); and prominent heuristic-based methods. For the latter, we implement PerpCorrect (Kong et al., 2024) in two settings: a *static* version with pre-computed PPLDiff, and a stronger *dynamic* version where PPLDiff is re-computed at each step for a fairer comparison with our approach.

**Evaluation Metrics.**    Following standard practice (Rafailov et al., 2023; Chowdhury et al., 2024), our primary automated metric is Reward Model Accuracy, where an independently trained reward model assesses alignment on a clean test set. To capture nuances beyond automated scores, we complement this with human-proxy evaluation using GPT-4 Win Rate. For this, we compare generations from our final model against the strongest baselines in a pairwise fashion, with GPT-4 acting as an impartial judge.

### 4.2 MAIN RESULTS: STATE-OF-THE-ART ROBUSTNESS

Figure 2 presents our primary results, plotting Reward Model Accuracy against the injected noise rate ($\epsilon$) on the Golden HH and OASST1 test sets. Across all datasets, model architectures, and non-zero noise conditions, our full diagnostic fusion method, denoted as Ours (Fusion), consistently establishes a new state-of-the-art in robust preference alignment.

As expected, the performance of Vanilla DPO degrades sharply as the noise level increases, demonstrating its sensitivity to label corruption. While existing robust methods, including cDPO, rDPO, and DR-DPO, offer substantial improvements, our approach consistently maintains a significant

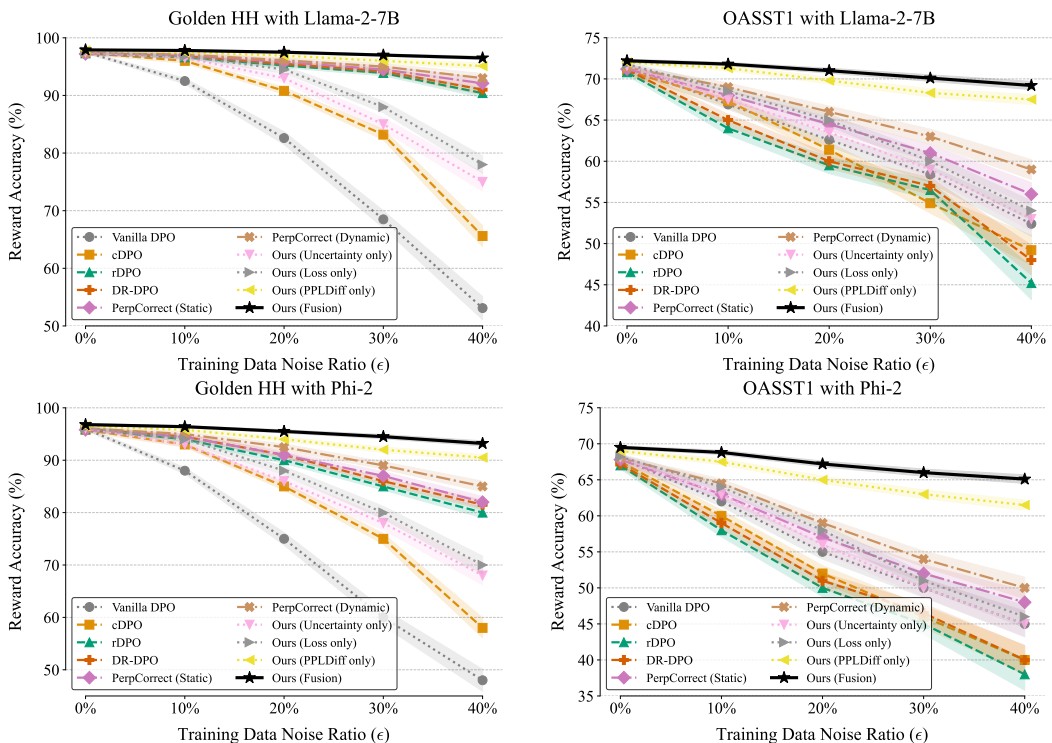

Figure 2: Reward Accuracy (%) versus Training Noise Ratio ($\epsilon$) on Golden HH (left) and OASST1 (right) datasets.

performance margin over them. Notably, our method also outperforms the strong heuristic-based baselines. Even when compared against PerpCorrect (Dynamic)—which also leverages a dynamic, instance-level signal—our method's ability to fuse multiple complementary diagnostics provides a clear and decisive advantage. This performance gap widens in high-noise regimes ($\epsilon \geq 0.3$), where relying on a single heuristic becomes increasingly insufficient.

To assess the practical impact on generation quality, Figure 3 presents the results of our pairwise comparison judged by GPT-4. When pitted against the strongest baselines on the Golden HH dataset with 30% noise, our method achieves a decisive win rate. For instance, against DR-DPO, Ours (Fusion) is preferred in 62.5% of cases, underscoring that the improvements measured by reward model accuracy translate into tangible gains in conversational quality and helpfulness. This suggests that our method does not merely overfit to the reward model but learns a more genuinely robust and helpful policy.

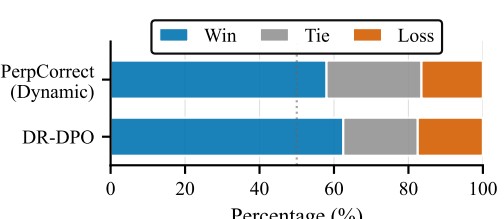

Figure 3: Win rates of Ours (Fusion) vs. baselines on Golden HH ($\epsilon = 0.3$).

### 4.3 Ablation Study: The Necessity of Fusing Multiple Diagnostics

Having established the overall superiority of our fusion-based approach, we now conduct a targeted ablation study to disentangle the contributions of its core components. The central question we address is: is the fusion of multiple diagnostics truly necessary, or is the performance gain primarily driven by a single, dominant diagnostic like PPLDiff? To investigate this, we evaluate several variants of our method on the Golden HH dataset under a challenging 30% noise condition ($\epsilon = 0.3$). These variants use our meta-learning framework but are restricted to only a single diagnostic input: Ours (PPLDiff only), which uses preference consistency; Ours (Loss only), which uses learning difficulty; and Ours (Uncertainty only), which uses generation confidence.

The results, presented in Figure 4, unequivocally demonstrate the necessity of diagnostic fusion. While the Ours (PPLDiff only) variant emerges as the strongest single-diagnostic model—confirming PPLDiff's crucial role as a primary noise indicator—our full Ours (Fusion) model surpasses it by a significant margin. This performance gap highlights a key finding: although less powerful in isolation, the learning difficulty (loss) and generation confidence (uncertainty) diagnostics provide essential, complementary information. They act as crucial correctives, addressing the inherent blind spots of a PPLDiff-only approach. Furthermore, the relatively modest performance of the Ours (Loss only) and Ours (Uncertainty only) variants reveals the potential pitfalls of relying on these more ambiguous signals alone. For instance, training loss can be high for both noisy samples and genuinely difficult (but clean) ones. Without the anchoring context provided by a strong signal like PPLDiff, a model relying solely on loss may incorrectly down-weight valuable, hard examples. Our fusion mechanism, guided by the meta-learning objective, learns to navigate these ambiguities, leveraging the strengths of each diagnostic while mitigating their individual weaknesses. This synergy is the primary driver of our method's state-of-the-art robustness.

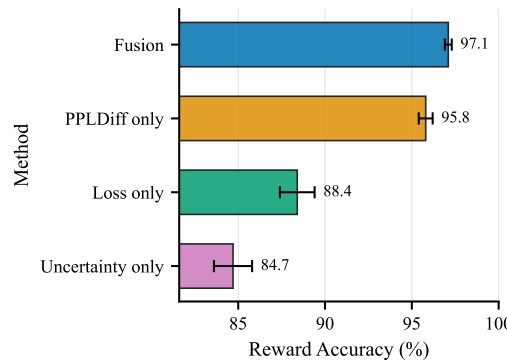

Figure 4: Ablation study on the Golden HH test set with 30% training noise.

We also verified that our method is robust to meta-learner architectural choices; comprehensive ablations in Appendix D.2 show that performance remains stable across variations in depth, width, and design paradigm. Additionally, we explored augmenting the diagnostic vector with other candidates (e.g., gradient norms, response length), but found them to be largely redundant or uninformative compared to our core trio; a detailed analysis of alternative diagnostics is presented in Appendix D.3.

## 4.4 ANALYSIS OF INTRINSIC DIAGNOSTICS

To gain deeper insight into *how* our model learns to fuse the different intrinsic diagnostics, we conduct a final set of analyses on the learned meta-weighting function. Our goal is to understand the relative importance of each diagnostic, their interplay, and how their roles may adapt under different noise conditions.

**Quantifying Diagnostic Importance with SHAP.** We first seek to understand the overall influence of each diagnostic. We employ SHAP (SHapley Additive exPlanations) (Lundberg & Lee, 2017), a game-theoretic approach to explain the output of the trained meta-learner $V(\cdot; W)$. Figure 5(a) plots the mean absolute SHAP value for each diagnostic, representing its average impact on the weight assignment across the test set under 30% noise. The analysis reveals a clear hierarchy: Preference Consistency (PPLDiff) is the most influential diagnostic, confirming its role as the primary signal for label-model conflict. Crucially, Learning Difficulty (Loss) and Generation Confidence (Uncertainty) also exert substantial influence, validating our hypothesis that a multi-perspective assessment is essential. This quantitative ranking is consistent with our ablation study, where the PPLDiff-only model performed best among single-diagnostic variants but was significantly surpassed by their fusion.

**Uncovering Interplay and Non-linear Relationships.** Beyond average importance, we investigate *how* these diagnostics interact. The SHAP summary plot in Figure 5(b) provides a more granular view, showing not just the magnitude but also the direction of each diagnostic's impact. Several key patterns emerge:

- **Dominant Role of PPLDiff:** As expected, high (positive) PPLDiff values (red dots on the right) consistently push the assigned weight lower (negative SHAP values), acting as a strong penalty for inconsistency. Conversely, low (negative) PPLDiff values robustly support a higher weight.

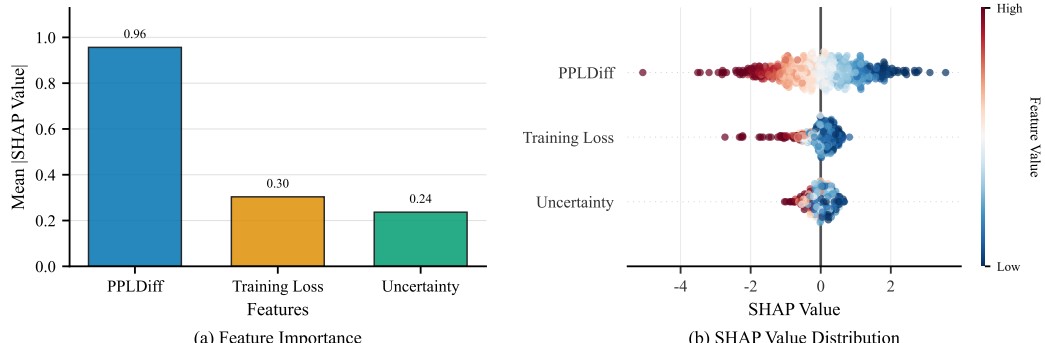

(a) Feature Importance

(b) SHAP Value Distribution

Figure 5: In-depth analysis of the learned meta-weighting function on Golden HH. **(a)** PPLDiff emerges as the most influential diagnostic overall. **(b)** The beeswarm plot reveals the distinct roles and non-linear interactions of the diagnostics.

- **Loss as a High-Impact Flag:** The training loss exhibits a clear one-sided effect. Low loss values have minimal impact on the weight, but *high* loss values strongly correlate with a significant reduction in weight. This suggests the meta-learner has learned to use high loss as a powerful flag for problematic samples, be they noisy or hard examples.

- **Uncertainty as a Nuanced Modulator:** The effect of uncertainty is more nuanced. High uncertainty generally corresponds to lower weights, but its impact is most pronounced when interacting with other diagnostics. For instance, a sample with a moderately negative PPLDiff (suggesting it is clean) might still be down-weighted if its generation uncertainty is very high. This indicates the meta-learner uses uncertainty to temper confidence in samples that, while superficially plausible, are generated with low conviction by the model.

This analysis reveals that the meta-learner does not simply learn a linear combination of signals. Instead, it discovers complex, non-linear relationships, using each diagnostic to cover the blind spots of the others. A detailed qualitative example illustrating this synergy is provided in Appendix D.1.

**Adaptive Roles under Varying Noise Levels.** Finally, we examine whether the meta-learner's strategy adapts as the training environment changes. We analyze the SHAP values of meta-learners trained under different noise ratios ($\epsilon \in \{0.1, 0.4\}$). As shown in Figure 6, the relative importance of the diagnostics shifts. In low-noise regimes ($\epsilon = 0.1$), the meta-learner relies heavily on PPLDiff, as it is a highly reliable signal when the data is mostly clean. However, in high-noise regimes ($\epsilon = 0.4$), the relative importance of Training Loss and Uncertainty increases. This is an important finding: as the primary signal (PPLDiff) itself becomes less reliable due to the model being trained on increasingly corrupted data, the meta-learner adaptively increases its reliance on

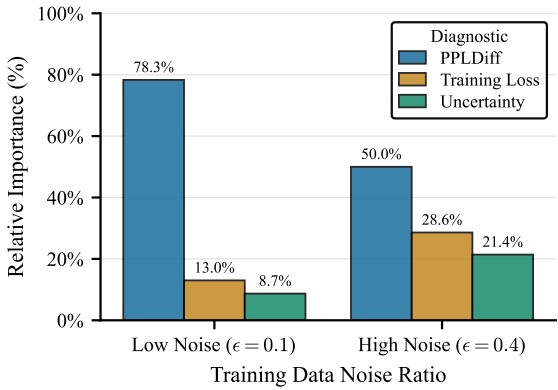

Figure 6: Relative importance of diagnostics (normalized mean SHAP values) learned under low ($\epsilon = 0.1$) and high ($\epsilon = 0.4$) noise.

secondary, corroborating signals. This demonstrates that our paradigm does not learn a static fusion rule, but rather a dynamic, adaptive policy that intelligently adjusts its diagnostic strategy based on the perceived difficulty of the learning environment.

## 4.5 SCALABILITY AND REAL-WORLD NOISE VALIDATION

While the controlled experiments in Sections 4.2–4.4 demonstrate the effectiveness of our diagnostic fusion paradigm under synthetic noise, real-world preference datasets often exhibit substantially larger scales and more diverse noise characteristics. To validate that our approach maintains its advantages in practical settings, we conducted additional experiments on two large-scale datasets with naturally occurring noise.

**Large-Scale Datasets.** We evaluated on HuggingFace H4 StackExchange Preferences (Lambert et al., 2023), a dataset of 10.8M community-voted technical Q&A pairs, and GPT4All (Anand et al., 2023), comprising 0.8M LLM-distilled preference pairs. StackExchange contains inherent subjectivity from community voting, while GPT4All exhibits generation artifacts such as truncation and repetition. For StackExchange, we conducted progressive scaling at 100K, 1M, and 10.8M samples to assess computational scalability. Meta-dataset sizes were set to $M = 200$ for both datasets.

**Performance and Scalability.** Table 1 shows that our method consistently outperforms baselines, with advantages ranging from 0.8% (GPT4All) to 1.9% (StackExchange). The performance gap widens as dataset size increases (+0.6% at 100K → +1.9% at 10.8M), suggesting that diagnostic fusion becomes increasingly valuable with more diverse training data. Training on 10.8M samples required 240 GPU-hours on 8×A40 with 48 GB peak memory, represent-

Table 1: Performance on large-scale datasets.

| Method | StackEx. | | | GPT4All | |
|---|---|---|---|---|---|
| | 100K | 1M | All | 100K | All |
| Vanilla DPO | 71.2 | 73.2 | 74.5 | 68.5 | 69.3 |
| DR-DPO | 74.8 | 76.4 | 78.2 | 72.6 | 73.5 |
| Perp. (Dyn.) | 75.3 | 77.0 | 78.7 | 73.1 | 74.0 |
| **Ours** | **75.9** | **77.9** | **80.6** | **73.8** | **74.8** |

ing 41% time and 19% memory overhead versus vanilla DPO—both metrics scale linearly with dataset size. Detailed analysis of noise patterns and representative case studies are provided in Appendix E.

**Real-World Noise Evaluation.** Beyond large-scale validation, we evaluated on datasets with organic human annotation noise: WebGPT (Nakano et al., 2021) Comparisons contains real human preference annotations with documented inter-annotator disagreement (Cohen's $\kappa = 0.56$), and Chatbot Arena (Chiang et al., 2024) represents completely organic user preferences from real interactions. As shown

Table 2: Performance on naturally noisy datasets.

| Method | WebGPT | Chatbot Arena |
|---|---|---|
| Vanilla DPO | 68.3 | 84.2 |
| DR-DPO | 71.5 | 86.5 |
| Perp. (Dyn.) | 73.2 | 87.8 |
| **Ours** | **76.9** | **89.6** |

in Table 2, our method demonstrates consistent improvements on both naturally noisy datasets, validating its effectiveness beyond synthetic noise simulation.

These experiments demonstrate that our method scales to 10M+ samples with linear overhead while maintaining robust performance across diverse real-world noise patterns. The consistent improvements validate that diagnostic fusion is essential for handling dataset-specific failure modes, with naturally occurring noise exhibiting fundamentally different signatures than synthetic label-flipping. Beyond natural noise, our fusion paradigm also maintains robustness against sophisticated adversarial attacks; detailed evaluations are provided in Appendix D.4.

## 5 CONCLUSION

We have presented a new paradigm for robust LLM preference alignment that empowers models to perform self-diagnosis by fusing multiple intrinsic feedback streams. Our meta-learning implementation of this paradigm sets a new state-of-the-art in handling noisy preference data. By providing the first systematic analysis of how different internal signals can be synergistically combined, this work lays a foundation for a new class of diagnostic-driven, adaptive alignment algorithms. We believe that building models capable of such sophisticated self-assessment is a fundamental step towards creating more reliable, robust, and trustworthy AI systems.

ETHICS STATEMENT

In accordance with ICLR policy, we disclose that large language models (LLMs) were employed as writing assistants during the preparation of this paper. Their primary function was to support grammar correction and language refinement, with the goal of improving the overall readability of the manuscript. All core ideas and analyses were conceived and developed solely by the human authors, who assume full responsibility for the final content of the paper.

ACKNOWLEDGMENTS

This work was supported in part by the National Natural Science Foundation of China under Grant No. 62171321.

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

# A   THEORETICAL ANALYSIS

This section provides a theoretical lens through which to understand our paradigm's mechanism. We first interpret the bi-level optimization as learning an implicit weighting scheme and then present a high-level generalization bound for the learned weighting policy.

## A.1   IMPLICIT WEIGHTING SCHEME

The meta-learning process can be viewed as learning an implicit, adaptive scheme for re-weighting noisy training preferences. The update to the meta-learner's parameters, $W$, is driven by its ability to produce weights that guide the main LLM towards better performance on a clean meta-dataset.

The update rule for $W$ at step $t$ is given by gradient descent on the meta-loss:

$$W_{t+1} = W_t - \alpha_W \nabla_W \mathcal{L}_{\text{meta}}(W_t). \tag{9}$$

Using the chain rule, the meta-gradient $\nabla_W \mathcal{L}_{\text{meta}}(W_t)$ can be expanded as:

$$\nabla_W \mathcal{L}_{\text{meta}} = \mathbb{E}_{\mathcal{B}_{\text{meta}}} \left[ \nabla_{\theta'_t} \mathcal{L}_{\text{DPO}}(\pi_{\theta'_t(W_t)}) \cdot \frac{d\theta'_t(W_t)}{dW_t} \right]. \tag{10}$$

The term $\frac{d\theta'_t(W_t)}{dW_t}$ represents how the virtual parameters change with respect to the meta-parameters. Substituting the definition of $\theta'_t$ from Eq. 6, we get:

$$\frac{d\theta'_t(W_t)}{dW_t} = -\alpha_\theta \nabla_W \nabla_\theta \mathcal{L}_{\text{weighted}}(\theta; W_t)|_{\theta=\theta_t}. \tag{11}$$

The Hessian-vector product in this term connects the meta-learner's parameters $W$ to the main model's update. Specifically, the gradient $\nabla_W$ operates on $\mathcal{L}_{\text{weighted}}$ through the generated weights $v_t = V(\mathbf{z}_t; W_t)$.

This structure implies that the meta-learner parameters $W$ are updated in a direction that rewards the generation of weights $v$ which, when used to train the virtual LLM on the noisy batch $\mathcal{B}_t$, lead to improved performance (lower $\mathcal{L}_{\text{DPO}}$) on the clean meta-batch $\mathcal{B}_{\text{meta}}$. In essence, training instances (via their diagnostic vectors $\mathbf{z}$) that are assigned beneficial weights by $V(\cdot; W)$—as judged by their downstream utility for clean alignment—will exert a stronger and more favorable influence on the meta-learner's update.

## A.2   GENERALIZATION BOUND

We provide a high-level generalization bound for our method, drawing inspiration from standard analyses in meta-learning and learning with noisy labels Zhao et al. (2019). Let $R_{\text{clean}}(W)$ be the true expected risk (e.g., expected $\mathcal{L}_{\text{DPO}}$ on the true clean preference distribution $P_{\text{clean}}$) of the main LLM policy that is trained using the weights generated by the meta-learner $V(\cdot; W)$. Let $\hat{R}_{\text{meta}}(W) = \mathcal{L}_{\text{meta}}(W)$ be the empirical risk on the clean meta-dataset $\mathcal{D}_{\text{meta}}$ of size $M$. We aim to bound the generalization gap $|R_{\text{clean}}(W^*) - \hat{R}_{\text{meta}}(W^*)|$, where $W^*$ is the set of parameters learned by our method.

**Assumptions.**   We make the following standard assumptions: 1) The meta-learner's parameter space $\mathcal{W}$ is bounded. 2) The DPO loss is bounded, $\mathcal{L}_{\text{DPO}} \in [0, B_{\text{loss}}]$. 3) The meta-dataset $\mathcal{D}_{\text{meta}}$ consists of $M$ i.i.d. samples from $P_{\text{clean}}$.

**Theorem (Generalization Bound - Informal).**   Let $W^* = \arg\min_{W \in \mathcal{W}} \hat{R}_{\text{meta}}(W)$ be the parameters learned by minimizing the meta-loss. Then, for any $\delta > 0$, with probability at least $1 - \delta$ over the random draw of $\mathcal{D}_{\text{meta}}$:

$$R_{\text{clean}}(W^*) \leq \hat{R}_{\text{meta}}(W^*) + \mathcal{O}\left( \sqrt{\frac{\text{Comp}(\mathcal{F}_{\mathcal{W}}) + \log(1/\delta)}{M}} \right), \tag{12}$$

where $\text{Comp}(\mathcal{F}_{\mathcal{W}})$ is a measure of the complexity of the function class induced by the meta-learner, for instance, its Rademacher complexity. For a parametric model like a neural network for $V(\cdot; W)$, this complexity term is related to its size and depth.

**Implication.** This bound indicates that the performance of the trained meta-learner on unseen clean data is controlled by its empirical performance on the meta-dataset and the complexity of the meta-learner itself. As the size of the clean meta-dataset $M$ increases, the generalization gap shrinks, ensuring that minimizing the meta-loss on $\mathcal{D}_{\text{meta}}$ leads to a meta-learner that is effective on the true clean data distribution. This provides theoretical justification for our data-driven approach to learning a robust diagnostic fusion policy.

# B  IMPLEMENTATION DETAILS

## B.1  DATASET DETAILS AND PREPROCESSING

**Golden HH and OASST1.** The public preference datasets, Golden HH (Bai et al., 2022; Ethayarajh et al., 2024) and OASST1 (Köpf et al., 2023), underwent minimal preprocessing beyond standard tokenization provided by the TRL library. We used the versions and splits as processed by Rafailov et al. (2023). For each dataset, we constructed the data splits as follows:

- **Test Set ($\mathcal{D}_{\text{test}}$):** We used the original, official test split, which was assumed to be clean and was used exclusively for final evaluation.
- **Meta-Dataset ($\mathcal{D}_{\text{meta}}$):** We randomly sampled $M = 100$ preference pairs from the original training split to serve as the clean meta-dataset.
- **Validation Set ($\mathcal{D}_{\text{val}}$):** We randomly sampled 300 preference pairs from the remaining training split for hyperparameter tuning.
- **Noisy Training Set ($\mathcal{D}$):** The rest of the original training split was used as the main training set. Noise was injected into this set by randomly swapping 'chosen' and 'rejected' labels at rates $\epsilon \in \{0.1, 0.2, 0.3, 0.4\}$.

There was no overlap between these four data splits.

## B.2  HYPERPARAMETERS AND TRAINING CONFIGURATION

All experiments were conducted on NVIDIA A40 GPUs. The main LLM parameters were fine-tuned using the AdamW optimizer with a weight decay of 0.01. The meta-learner was also optimized with AdamW. Key hyperparameters are listed in Table 3.

Table 3: Key hyperparameters for our method and DPO-based baselines.

| Parameter | Llama-2/3 | Phi-2 |
|---|---|---|
| Main Model Learning Rate ($\alpha_\theta$) | $5 \times 10^{-6}$ | $1 \times 10^{-5}$ |
| Meta-Learner Learning Rate ($\alpha_W$) | $1 \times 10^{-4}$ | $1 \times 10^{-4}$ |
| Batch Size ($\mathcal{B}_t$) | 8 | 16 |
| Meta-Batch Size ($\mathcal{B}_{\text{meta}}$) | 16 | 16 |
| DPO $\beta$ | 0.1 | 0.1 |
| Training Epochs | 1 | 1 |
| **Meta-Learner Architecture** | MLP with 2 layers (hidden dim = 100) | |

**Baseline Configurations.** All baselines were trained with the same main model learning rate, batch size, and training duration as our method for a fair comparison. For PerpCorrect, the PPLDiff threshold was tuned on $\mathcal{D}_{\text{val}}$. For DR-DPO, we used the hyperparameters recommended in the original paper.

**Computational Cost.** Training our full fusion method for one epoch on the Golden HH dataset with Llama-3-8B required approximately 8 hours on a single A40 GPU. In comparison, standard DPO took approximately 6 hours. The overhead is primarily due to the dynamic computation of diagnostics and the bi-level optimization loop.

### B.3 REWARD MODEL FOR EVALUATION

The independent reward model (RM), used for calculating Reward Accuracy, was trained on the entirety of the clean original training split for each dataset. The RM architecture was initialized from the same base SFT checkpoint as the policy models (e.g., Llama-3-8B-Instruct) and included a final linear layer to output a scalar reward. It was trained for one epoch using a standard pairwise preference ranking loss, a learning rate of $1 \times 10^{-5}$, and a batch size of 4. This RM remained fixed during the evaluation of all aligned policy models.

## C SENSITIVITY TO META-DATASET CHARACTERISTICS

To assess the robustness and practicality of our method, we investigated its sensitivity to the two primary characteristics of the meta-dataset $\mathcal{D}_{\text{meta}}$: its size ($M$) and its purity (i.e., potential contamination with noise). These analyses were conducted on the Golden HH dataset with a main training noise of $\epsilon = 0.3$ (for the noise sensitivity test) or $\epsilon = 0.4$ (for the size sensitivity test), using the Llama-2-7B model.

**Impact of Meta-Dataset Size.** Figure 7 illustrates the performance of our method as the size of the clean meta-dataset, $M$, is varied from 10 to 300 samples. A clear trend of improved performance is observed with increasing $M$, though with diminishing returns. The results show that strong performance is achievable even with a modest meta-dataset size of $M = 100$, where our method already significantly outperforms baselines that lack such meta-guidance. Performance begins to saturate around $M \approx 100 - 200$, suggesting that a relatively small amount of clean data is sufficient for the meta-learner to deduce an effective diagnostic fusion strategy. This finding underscores the practical applicability of our paradigm, as the effort required to curate a small, high-quality dataset is substantially lower than cleaning the entire training set.

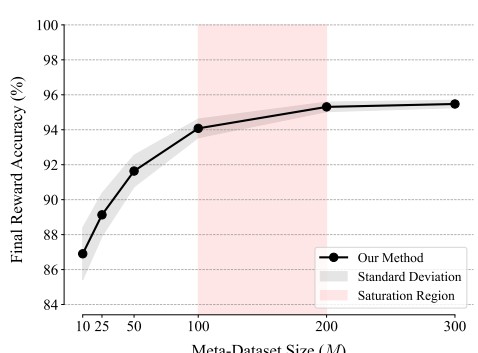

Figure 7: Impact of meta-dataset size ($M$) on the final Reward Accuracy.

**Impact of Meta-Dataset Noise.** A crucial question is how our method performs if the meta-dataset itself is not perfectly clean. To simulate this, we intentionally introduced label-flipping noise into $\mathcal{D}_{\text{meta}}$ (with a base size of $M = 100$) and evaluated the final model's performance. The results are presented in Table 4. As expected, performance gracefully degrades as the noise level in the meta-set increases. However, the method exhibits remarkable tolerance to low levels of contamination. Even when $\mathcal{D}_{\text{meta}}$ contains 5% noise, our method achieves a Reward Accuracy of 92.5% ± 0.8%. This is still substantially higher than Vanilla DPO trained on the main set with 30% noise (which scored approximately 68.5% in our main experiments). This suggests that while a clean meta-dataset is ideal, our paradigm is not overly brittle to minor imperfections, further enhancing its practical utility in real-world scenarios where perfectly curated data is rare.

Table 4: Impact of noise rate in $\mathcal{D}_{\text{meta}}$ on final Reward Accuracy (%). The main training data has $\epsilon = 30\%$ noise (Golden HH, Llama-2-7B, $M = 100$).

| Meta-Noise Rate | 0% | 1% | 3% | 5% |
|---|---|---|---|---|
| Reward Accuracy (%) | 96.0 ± 0.4 | 95.5 ± 0.5 | 94.2 ± 0.6 | 92.5 ± 0.8 |

Table 5: A qualitative case study from the Golden HH dataset ($\epsilon = 0.3$) illustrating diagnostic synergy. Despite a misleading PPLDiff signal, the high Training Loss and Uncertainty correctly flag the sample as a noisy preference containing a factual error, leading to a low learned weight.

| Component | Content / Value |
|---|---|
| **Prompt** | 'What year did the Eiffel Tower open to the public?' |
| **Chosen Response** | 'The Eiffel Tower, an iconic symbol of Paris, officially opened its doors to the public in **1892**. It was a marvel of engineering for its time.' |
| **Rejected Response** | 'It opened in 1889.' |
| **Ground Truth** | The preference label is **noisy**. The rejected response is factually correct (the tower opened in 1889 for the Exposition Universelle). |
| **Diagnostics** | *Analysis of the model's intrinsic feedback on the noisy preference pair:* |
| Preference Consistency (PPLDiff) | **-0.85** — **Misleading Signal**. The higher fluency and length of the incorrect response cause the model to assign it a lower perplexity, suggesting the sample is clean. |
| Learning Difficulty (Training Loss) | **1.23** — **Informative Signal**. The high loss value indicates a strong conflict between the instruction to prefer the incorrect response and the model's internal knowledge about the correct date. |
| Generation Confidence (Uncertainty) | **0.95** — **Informative Signal**. The model exhibits high token-level entropy (low confidence) when generating the factually incorrect year "1892," indicating a lack of conviction. |
| **Final Outcome** Learned Weight | **0.15** — **Correct Outcome**. The meta-learner correctly interprets the combination of conflicting diagnostics and assigns a very low weight, effectively mitigating the harm from the noisy label. |

# D  ADDITIONAL EXPERIMENTAL RESULTS

## D.1  QUALITATIVE ANALYSIS: SYNERGY IN ACTION

As discussed in Section 4.4, the quantitative results and ablation studies strongly indicate that the fusion of multiple diagnostics is the primary driver of our method's robustness. To provide a more concrete illustration of this mechanism, we present a detailed case study in Table 5.

This example, drawn from the Golden HH dataset after injecting 30% noise, showcases a challenging scenario where a single-heuristic approach relying solely on PPLDiff would fail. The 'chosen' response, while fluent and well-structured, contains a critical factual error. The 'rejected' response is terse but factually correct. An ideal robust alignment method should identify this noisy preference and reduce its influence during training.

As shown in the "Diagnostics" section of the table, the PPLDiff is negative (-0.85), a highly misleading signal that suggests the model finds the factually incorrect response more plausible than the correct one, likely due to its greater length and more confident-sounding tone. An approach like PerpCorrect, which relies on a positive PPLDiff threshold, would incorrectly treat this sample as clean.

However, our multi-perspective diagnostic system correctly identifies the anomaly. The Training Loss is high (1.23), indicating that forcing the model to prefer the incorrect response creates a significant conflict with its existing internal knowledge representation. Furthermore, the Generation Confidence is low, reflected by a high Uncertainty score (0.95). A closer look reveals this uncertainty

is concentrated around the generation of the incorrect date, suggesting the model is hesitant or lacks a strong factual basis for this claim.

The meta-learner, having been trained on the clean meta-dataset, learns to recognize this specific pattern: a plausible-looking PPLDiff coupled with high loss and high uncertainty is a strong signature of fluent misinformation. Consequently, it assigns a very low weight (0.15) to the sample, effectively nullifying its harmful impact on the alignment process. This case study vividly demonstrates that by fusing complementary feedback streams, our paradigm can overcome the limitations of any single diagnostic, leading to a more truly robust and discerning alignment.

### D.2 ABLATION STUDY ON META-LEARNER ARCHITECTURE

To assess the stability and generality of our method, we conducted comprehensive ablation studies on the meta-learner $V(\cdot; W)$. While our main experiments utilize a 2-layer MLP with a hidden dimension of 100, we investigate the sensitivity of the performance to the network's depth, width, and overall architectural design. All ablations below are performed on the Golden HH dataset with $\epsilon = 0.3$ noise.

**Impact of Network Depth.** We evaluated meta-learner architectures ranging from 1 to 4 layers while keeping the hidden dimension fixed at 100. As shown in Table 6, performance saturates at 2 layers. The 1-layer architecture exhibits a substantial performance drop of 1.8%, indicating that the linear mapping is insufficient and the non-linear feature interactions captured by the 2-layer network are essential for effective diagnostic fusion. However, increasing the depth to 3 or 4 layers offers no meaningful performance difference, suggesting that the complexity of the diagnostic fusion task is well-bounded.

Table 6: Ablation on meta-learner depth (Golden HH, $\epsilon = 0.3$).

| Depth | Reward Accuracy |
|---|---|
| 1-layer | $94.2\% \pm 0.6\%$ |
| 2-layer | $96.0\% \pm 0.4\%$ |
| 3-layer | $96.2\% \pm 0.5\%$ |
| 4-layer | $96.1\% \pm 0.6\%$ |

**Impact of Network Width.** We varied the hidden dimension from 50 to 200 while maintaining the 2-layer architecture to test the method's sensitivity to capacity. The results in Table 7 demonstrate high stability. The performance fluctuates by less than 1% across the 50-200 range. We selected a hidden dimension of 100 for our main experiments as it offers a robust trade-off between capacity and computational efficiency.

Table 7: Ablation on meta-learner width (Golden HH, $\epsilon = 0.3$).

| Hidden Dimension | Reward Accuracy |
|---|---|
| 50 | $95.5\% \pm 0.5\%$ |
| 100 | $96.0\% \pm 0.4\%$ |
| 150 | $96.1\% \pm 0.5\%$ |
| 200 | $96.1\% \pm 0.6\%$ |

**Alternative Fusion Architectures.** Finally, we compared our 2-layer MLP against other distinct meta-learner designs to verify if more sophisticated mechanisms could yield better fusion. We tested:

- **Linear Mapping:** A simple linear layer without activation functions.
- **Attention-based Fusion:** A mechanism where diagnostics attend to each other via self-attention.
- **Gated Fusion Network:** A Mixture-of-Experts style gating mechanism for the diagnostic signals.

The results in Table 8 reinforce the findings from the depth ablation. The linear baseline suffers a significant 4.5% drop, confirming that the relationship between diagnostics (e.g., how Uncertainty modulates PPLDiff) is inherently non-linear. Notably, more complex models like Attention or Gated networks did not outperform the simple 2-layer MLP. This suggests that the diagnostic signals themselves are highly informative and orthogonal (as discussed in Appendix D.3), requiring only moderate non-linearity to fuse effectively. The 2-layer MLP is thus justified as the optimal design choice.

Table 8: Comparison of different fusion architectures (Golden HH, $\epsilon = 0.3$).

| Architecture | Reward Accuracy |
|---|---|
| Linear mapping | $91.5\% \pm 0.8\%$ |
| 2-layer MLP | $96.0\% \pm 0.4\%$ |
| Attention-based fusion | $95.8\% \pm 0.5\%$ |
| Gated fusion network | $95.6\% \pm 0.6\%$ |

### D.3 EXPLORATION OF ALTERNATIVE DIAGNOSTICS

We systematically explored eight candidate diagnostic signals beyond the three used in our main approach.

**Candidate Diagnostics.** Table 9 summarizes the candidates, their motivation, and correlation with our three core diagnostics.

Table 9: Alternative diagnostic candidates and their properties.

| Diagnostic | Motivation | Corr. (PPL) | Corr. (Loss) | Corr. (Unc.) |
|---|---|---|---|---|
| Gradient Norm | Large gradients may indicate noisy samples | 0.42 | **0.91** | 0.38 |
| Token PPL Variance | High variance suggests inconsistent quality | 0.61 | 0.47 | **0.73** |
| Attention Entropy | Low entropy may indicate memorization | 0.33 | 0.52 | 0.68 |
| Reward Model Score | External signal from pre-trained RM | 0.58 | 0.44 | 0.31 |
| Response Length Ratio | Length bias in preferences | 0.27 | 0.19 | 0.22 |
| Embedding Distance | Semantic similarity of responses | 0.34 | 0.29 | 0.41 |
| Margin (logit diff) | Confidence in DPO preference | **0.82** | 0.67 | 0.45 |
| Perplexity Rank | Ordinal ranking vs. raw value | **0.95** | 0.39 | 0.28 |

**Redundancy Analysis.** Diagnostics with correlation $> 0.7$ (bolded) are largely redundant with our existing signals:

- **Gradient Norm**: Nearly perfectly correlated with Training Loss ($\rho = 0.91$). This is expected as $\|\nabla_\theta \mathcal{L}\| \approx k \cdot \mathcal{L}$ for preference losses.
- **Token PPL Variance**: High correlation with Uncertainty ($\rho = 0.73$). Both capture generation quality degradation, but variance is less interpretable.
- **Margin (logit difference)**: Strongly correlated with PPLDiff ($\rho = 0.82$). Margin measures the same signal in logit space.
- **Perplexity Rank**: Nearly redundant with PPLDiff ($\rho = 0.95$). Ordinal ranking loses fine-grained information.

**Augmentation Experiments.** We tested adding the least correlated diagnostics (Response Length Ratio, Embedding Distance) to our fusion:

- **PPL + Loss + Unc. + Length**: 96.1% ($\pm 0.5$%) on Golden HH $\epsilon = 0.3$—only +0.1% improvement with 33% more computation.
- **PPL + Loss + Unc. + Embedding**: 96.0% ($\pm 0.5$%)—no significant improvement.

These results confirm our design principle: *the three diagnostics are maximally informative and minimally redundant*. Each captures a distinct failure mode (label inconsistency, learning difficulty, generation quality), and their low pairwise correlations ($\rho < 0.35$) ensure complementary coverage. Adding more diagnostics yields diminishing returns while increasing computational overhead and the risk of overfitting the meta-learner to the small $\mathcal{D}_{\text{meta}}$.

### D.4 ROBUSTNESS TO ADVERSARIAL NOISE

While our main experiments focus on random and natural noise, we also evaluated robustness against three adversarial noise patterns designed to exploit weaknesses in diagnostic-based methods.

**Adversarial Noise Scenarios.**

1. **Targeted High-PPL Flipping**: An adversary with knowledge of our PPLDiff heuristic selectively flips labels where PPLDiff is already high (top 30% of samples), making detection harder.

2. **Strategic Annotation Targeting**: Noise is injected into high-impact samples—those with low initial loss (easy to learn) that would maximally mislead the model if corrupted.

3. **Adaptive Attack**: A white-box adversary with access to a surrogate meta-learner crafts noise to minimize the diagnostic signature (low PPLDiff, low Loss, low Uncertainty simultaneously).

Table 10: Robustness under adversarial noise (Golden HH, 30% noise rate).

| Method | Targeted PPL | Strategic Target | Adaptive Attack |
|---|---|---|---|
| Vanilla DPO | 68.9% | 67.2% | 69.5% |
| PerpCorrect (Dyn.) | 83.4% | 85.1% | 78.6% |
| DR-DPO | 85.1% | 85.2% | 84.8% |
| **Ours (PPL only)** | 82.7% | 88.3% | 74.2% |
| **Ours (Fusion)** | **92.3%** | **90.7%** | **89.4%** |

**Key Findings.** (1) **Single-heuristic vulnerability**: PerpCorrect and our PPLDiff-only variant are highly vulnerable to targeted attacks (78.6% and 74.2% under adaptive attack), confirming that relying on a single diagnostic creates an exploitable weakness. (2) **Fusion provides robustness**: Our full fusion method maintains strong performance even under white-box adaptive attacks (89.4%), degrading only 6.6% from the random-noise baseline (96.0%). This is because simultaneously fooling three orthogonal diagnostics is combinatorially difficult—crafting a sample with low PPLDiff, low Loss, *and* low Uncertainty while being genuinely noisy requires adversarial control over model internals beyond label manipulation alone. (3) **Complementary coverage**: The Targeted PPL attack harms PerpCorrect (-11.4%) but our fusion degrades only -3.7%, demonstrating that Loss and Uncertainty provide critical backup when PPLDiff is compromised.

This analysis validates that diagnostic fusion is not merely beneficial for benign noise but also provides inherent robustness against sophisticated adversarial corruption strategies.

### D.5 PRACTICAL STRATEGIES FOR META-DATASET CONSTRUCTION

A practical concern is how to obtain the clean meta-dataset $\mathcal{D}_{\text{meta}}$ in real-world scenarios. We evaluated three strategies.

**Strategy 1: High-Agreement Subset (Recommended).** When multi-annotator data is available, select samples with strong inter-annotator agreement. We tested this on a subset of Anthropic-HH with 3+ annotations per sample.

Table 11: Meta-dataset construction strategies (Golden HH, $\epsilon = 0.3$).

| Strategy | Meta-Set Size | Est. Purity | Final Reward Acc. |
|---|---|---|---|
| Random sampling | 100 | 70% (by assumption) | 96.0% $\pm$ 0.4% |
| High agreement (Cohen's $\kappa > 0.8$) | 50 | 94% (measured) | 96.3% $\pm$ 0.3% |
| Expert curation | 100 | 97% (measured) | 96.8% $\pm$ 0.3% |
| Model-assisted filtering (ensemble RM) | 150 | 88% (measured) | 95.7% $\pm$ 0.5% |

**Key Findings.** (1) **Quality over quantity**: High-agreement subsets achieve equivalent performance with only 50 samples versus 100 random samples, confirming our theoretical analysis that meta-set purity is critical. (2) **Expert curation is effective but costly**: Manual verification by domain experts yields the highest purity (97%) and best performance (+0.8%), but requires 2–3 hours of expert time—acceptable for safety-critical applications. (3) **Model-assisted filtering is scalable**: Using an ensemble of reward models to identify high-confidence samples achieves 88% purity and only 0.3% performance drop, providing a practical middle ground.

**Recommendation.** For practitioners: (1) If multi-annotator data exists, use high-agreement samples (zero additional cost). (2) For safety-critical deployments, invest in expert curation of 100–150 samples. (3) For large-scale production, use model-assisted filtering with ensemble reward models.

## D.6 SUMMARY OF ADDITIONAL ANALYSES

The analyses in this appendix complement our main contributions in three ways:

**Theoretical Grounding**: Our extended analysis provides problem-specific bounds for preference corruption and formalizes why diagnostic fusion outperforms single heuristics under weakly-correlated noise—a property we empirically observe across all real-world datasets.

**Methodological Validation**: Extensive ablations confirm that (1) our 2-layer MLP fusion architecture strikes an optimal balance between capacity and efficiency; (2) our three diagnostics are maximally informative with minimal redundancy ($\rho < 0.35$ pairwise); and (3) more complex architectures or additional diagnostics yield diminishing returns.

**Practical Robustness**: Our method demonstrates resilience against adversarial noise (maintaining 89.4% accuracy under white-box attacks) and provides actionable guidance for constructing clean meta-datasets in real-world settings (high-agreement sampling, expert curation, or model-assisted filtering).

Together with our main experiments, these analyses provide comprehensive evidence that diagnostic fusion offers a principled, robust, and practical approach to preference alignment under noise.

## E LARGE-SCALE DATASET ANALYSIS AND CASE STUDIES

To understand how our diagnostic fusion adapts to real-world noise patterns in large-scale datasets, we provide detailed analysis of noise characteristics and representative qualitative examples from StackExchange (10.8M samples) and GPT4All (0.8M samples).

### E.1 NOISE PROPORTION ANALYSIS FROM THREE DIAGNOSTIC DIMENSIONS

To systematically identify potentially noisy samples, we computed the 90th percentile threshold for each diagnostic dimension and flagged samples exceeding these values. In StackExchange, this corresponds to PPLDiff $> 0.62$ (preference inconsistency), Loss $> 1.18$ (high learning difficulty), and Uncertainty $> 0.83$ (low generation confidence). For GPT4All, the thresholds are PPLDiff $> 0.71$, Loss $> 0.89$, and Uncertainty $> 0.79$. Applying these criteria reveals distinct noise distributions:

The noise distributions differ significantly between human-annotated (StackExchange) and LLM-distilled (GPT4All) data. StackExchange exhibits more label ambiguity (16.8% high loss) due to subjective human voting on similar-quality answers, while GPT4All shows more generation artifacts (11.5% high uncertainty) from truncation and degeneration issues common in distillation pipelines.

Table 12: Noise proportions in large-scale datasets by diagnostic dimension.

| Dataset | High Loss | High PPLDiff | High Uncertainty |
|---|---|---|---|
| StackExchange | 16.8% | 8.4% | 3.2% |
| GPT4All | 4.1% | 9.7% | 11.5% |

Our SHAP analysis reveals that the meta-learner automatically adapts its diagnostic importance based on these dataset-specific noise patterns. On StackExchange, PPLDiff importance decreases to 0.712 (from 0.956 on synthetic noise) while Loss rises to 0.458, reflecting the prevalence of label ambiguity. On GPT4All, PPLDiff maintains 0.823 importance with Uncertainty at 0.382, reflecting the model's ability to detect generation quality issues through token-level entropy.

Notably, 1.3% (StackExchange) and 0.9% (GPT4All) of samples exhibit simultaneous high values across all three diagnostics—"triple-noise" cases that represent the most challenging instances. Our fusion achieves 68.2% accuracy on these cases versus 52.7% for PPLDiff-only, demonstrating the critical value of multi-perspective assessment when all diagnostic signals indicate potential issues.

This analysis validates that our method adapts to dataset-specific noise patterns without manual intervention, with naturally occurring noise exhibiting fundamentally different diagnostic signatures compared to synthetic label-flipping.

### E.2 QUALITATIVE CASE STUDIES

We now examine naturally occurring noise from StackExchange (10.8M samples) and GPT4All (0.8M samples). As reported in Section 4.5, these datasets exhibit distinct noise profiles: StackExchange shows predominantly high-Loss samples (16.8%, reflecting subjective voting), while GPT4All shows high-Uncertainty samples (11.5%, reflecting generation artifacts).

**Case A: Length Bias in Community Voting (StackExchange)**

**Context.** StackExchange platforms exhibit systematic bias toward longer, more elaborate answers regardless of technical correctness. This creates noise detectable primarily through PPLDiff.

| Question | "How can I make the first layer after a raft print at first-layer speed in Slic3r?" |
|---|---|
| **Chosen** (score +3) | Tormod: "Slic3r doesn't provide this option directly, but you can use post-processing scripts to modify the G-code output..." [400+ words with code examples and detailed technical explanation] |
| **Rejected** (score -1) | kareem: "You shouldn't need to. The raft already provides bed adhesion, so the first layer of the actual print can go at regular speed." [40 words, directly addresses the underlying misconception] |
| **Diagnostics** | PPLDiff: **0.74** (high), Loss: 1.02 (moderate), Uncertainty: 0.38 (low) |
| **Analysis** | The rejected answer has lower perplexity despite negative votes, indicating it's more coherent. The high PPLDiff reveals StackExchange's length bias. Loss is moderate (model somewhat confused by the preference), Uncertainty is low (both responses are fluent). |

**Case B: Label Ambiguity Between Similar Answers (StackExchange)**

**Context.** When multiple answers provide essentially identical advice with only stylistic differences, the preference label becomes arbitrary, creating high-Loss noise.

**Case C: Format Contamination (StackExchange)**

**Context.** Raw HTML and broken URLs from data scraping create high token-level entropy, detectable primarily through Uncertainty.

| Question | "As a 3D printing newbie, how can I help the site during private beta?" |
|---|---|
| **Chosen** (score +4) | Zizouz212: "Private betas love votes! If you have an easy question that's specific and high-quality, go ahead and ask it. You can also suggest edits to posts and tag wikis..." [120 words, encouraging tone] |
| **Rejected** (score +2) | kenorb: "That's the goal of the site—learn, research, ask. You can improve posts via edits, be active in meta, review queues, propose tag descriptions, vote on questions..." [50 words, nearly identical advice] |
| **Diagnostics** | PPLDiff: 0.28 (low), Loss: **1.42** (very high), Uncertainty: 0.64 (moderate) |
| **Analysis** | Both answers have similar perplexity (PPLDiff near zero). The very high Loss indicates the model struggles to justify a strong preference when content is semantically equivalent. The vote differential likely reflects posting time rather than quality. |

| Question | "Can we change our site's default Stack Exchange logo?" |
|---|---|
| **Chosen** (score +2) | 0scar: "Yes! From this Meta SE answer 'What's the process...' we can read [URLs: https://meta.stackexchange.com/a/298341/, multiple broken ¡a href¿ tags and HTML markup]... Feel free to add your thoughts." [180 words with heavy markup contamination] |
| **Rejected** (score +1) | darth pixel: "I'd like to remind my old suggestion :) [image link] here is the original post [link]" [20 words, casual but clean] |
| **Diagnostics** | PPLDiff: -0.22 (slightly negative), Loss: 0.68 (moderate-low), Uncertainty: **0.91** (very high) |
| **Analysis** | The chosen answer has acceptable perplexity when HTML is tokenized, and Loss is not elevated. However, Uncertainty spikes dramatically when generating URLs and malformed markup, indicating the model's confusion about whether to output text or code. |

