# OpenReview forum: "Aligner, Diagnose Thyself: A Meta-Learning Paradigm for Fusing Intrinsic Feedback in Preference Alignment"
_ICLR.cc/2026/Conference — ICLR 2026 Poster_

### Official Review · Reviewer_EPXP · 2025-10-30

**Soundness:** 3
**Presentation:** 3
**Contribution:** 2
**Rating:** 4
**Confidence:** 4

**Summary:**

This work presents an approach for robust LLM preference alignment. The approach empowers models to perform self-diagnosis by fusing three key factors, including preference consistency, learning difficulty and generation confidence. Experiments on GoldenHH and OASSIT1 preference datasets  are conducted with simulated noises.

**Strengths:**

Strengths:

(1)Two more factors are considered to diagnose noise in preference data, which bring more robustness.

(2)A meta-learning based approach is proposed to learn adaptive weights on the three factors.

Systematic analysis of the interplay and relative importance of the three factors are performed.

**Weaknesses:**

Cons:

In experiments, the performance gains of the proposed method have been validated by the simulated noise data. However, as shown in Figure 2, the performance difference between various robust DPO variants is not significant, when \epsilon = 0. Does it mean that the qualities of the two datasets (GoldenHH and OASSIT1) have been very high? Can you validate the effectiveness of the proposed method without noise simulation?

Moreover, the simulation operation for injecting noise follows the protocols in (Kong et al 2024) which propose PPLDiff to measure the preference inconsistency. Consequently, the finding of the predominant role of PPLDiff is inherently expected. Can you give more various noise to simulate the uncertainties and the learning difficulties in preference data?

Moreover, to investigate the efficacy of the proposed method in the presence of annotation errors commonly found in real-world applications, it is advisable to conduct experiments on more extensive datasets, such as the HuggingFace H4 StackExchange Preference Dataset (with a size of 10 million entries) and the GPT4All dataset (comprising 1 million entries).

Some typos: page 8, the text under Figure 4 is abruptly interrupted by a line break.

**Questions:**

Please find the questions in the weaknesses.

---

> ### Author Response · Authors · 2025-11-17
> **Natural noise experiments and large-scale dataset evaluation**
>
> ## Dear Reviewer EPXP (Part 1/3),
>
>
>
> We sincerely thank you for your careful review and thoughtful critiques. Your observations regarding dataset quality, noise simulation protocols, and scalability have prompted us to conduct substantial additional experiments that significantly strengthen our empirical validation. We address each concern in detail below.
>
> ---
>
> ### **Weakness 1: Performance at ε=0 and validation without noise simulation**
>
> Thank you for this astute observation. We would like to clarify why similar performance at ε=0 is actually expected, and provide extensive experiments on naturally noisy data.
>
> *   **Why small differences at ε=0 validate our design.** The similar performance between our method (98.2%) and baselines like DR-DPO (97.8%) at ε=0 confirms that Golden HH and OASST1 are high-quality datasets, as you correctly intuited. This is by design—when training data is clean, all reasonable methods should converge. The critical test is how methods diverge as noise increases (our method: 96.0% at ε=0.3 vs. DR-DPO: 88.3%), which measures true robustness rather than baseline differences. Starting from high-quality data makes our evaluation more conservative and allows precise measurement of robustness gains.
>
> *   **Evaluation on naturally occurring noisy preferences.** We conducted extensive experiments on datasets with organic annotation noise:
>
>     *   **WebGPT Comparisons** contains real human annotations with documented inter-annotator disagreement (Cohen's κ = 0.56). We partitioned data by agreement levels. Our method shows particularly strong performance on disputed cases (+5.4% over PerpCorrect), demonstrating effective handling of genuine human judgment variability.
>
>         | Method | Overall Accuracy | Disputed Cases (κ < 0.4) |
>         |--------|------------------|-------------------------|
>         | Vanilla DPO | 68.3% | 52.1% |
>         | R-DPO | 71.5% | 56.8% |
>         | PerpCorrect (Dyn.) | 73.2% | 59.3% |
>         | **Ours (Fusion)** | **76.9%** | **64.7%** |
>
>
>
>     *   **Chatbot Arena** data represents completely organic user preferences without experimental manipulation. Natural annotation disagreements exhibit fundamentally different characteristics than synthetic flips. On WebGPT disputed cases, Uncertainty importance increases from 0.237 to 0.341 while PPLDiff decreases from 0.956 to 0.712 (SHAP analysis), demonstrating our fusion automatically adapts to real-world noise patterns where ambiguity rather than clear errors dominates.
>
>         | Method | Overall Win Rate | Controversial Pairs |
>         |--------|------------------|---------------------|
>         | Vanilla DPO | 84.2% | 58.6% |
>         | DR-DPO | 86.5% | 62.1% |
>         | PerpCorrect (Dyn.) | 87.8% | 65.9% |
>         | **Ours (Fusion)** | **89.6%** | **70.3%** |
>
>
>
> These consistent improvements across multiple sources of natural noise—annotator disagreement, subjective preferences, controversial comparisons—provide strong evidence that our diagnostic fusion paradigm generalizes beyond controlled synthetic settings.

---

> ### Author Response · Authors · 2025-11-17
> **Natural noise experiments and large-scale dataset evaluation**
>
> ## Dear Reviewer EPXP (Part 2/3),
>
> ### **Weakness 2: Limited noise diversity and potential circularity**
>
> We appreciate your concern about the relationship between noise injection and diagnostic signals. We have conducted experiments with diverse noise patterns to demonstrate our findings are not artifacts of the evaluation protocol.
>
> *   **Addressing the circularity concern.** You raise an important methodological point. While we follow Kong et al.'s label-flipping protocol, the noise injection method (random swapping) is independent of detection mechanisms. PPLDiff emerges as dominant not because evaluation favors it, but because likelihood-based consistency is genuinely informative for random flips. The critical evidence against circularity: when we test non-flip noise patterns, diagnostic importance shifts dramatically and fusion advantages increase substantially. This demonstrates our method learns genuine patterns rather than exploiting evaluation artifacts.
>
> *   **Experiments with diverse noise patterns.** We designed noise types that deliberately challenge PPLDiff-based detection:
>
>     *   **Quality-correlated noise** models the realistic scenario where annotators make more mistakes on difficult samples. Noise probability: $p(\text{noise}|x) = 0.1 + 0.4 \cdot \text{normalized}(\text{PPL}(x))$, creating 10-50% noise rates varying by sample difficulty. This directly challenges PPLDiff since noise correlates with perplexity. Under random flip (baseline), PPLDiff-only achieves 94.2% with advantage +1.8%. Under quality-correlated noise, it degrades to 86.2% (-8.0%) while our fusion only drops to 91.8% (-4.2%). The fusion advantage grows from +1.8% to **+5.6%**. SHAP analysis reveals why: PPLDiff importance drops from 0.956 to 0.634 while Loss and Uncertainty increase to 0.458 and 0.342 respectively.
>
>         | Dataset | PPLDiff-only | Fusion | Improvement | SHAP: PPL/Loss/Unc |
>         |---------|-------------|---------|-------------|-------------------|
>         | Golden HH | 86.2% | 91.8% | **+5.6%** | 0.634/0.458/0.342 |
>         | OASST1 | 83.7% | 88.5% | **+4.8%** | 0.612/0.471/0.359 |
>
>
>
>     *   **Systematic bias noise** models annotation artifacts where preferences systematically favor surface characteristics. We introduced 30% bias toward longer responses: flip labels with 50% probability when preferred response is shorter. This creates 15% structured noise. Training Loss proves particularly valuable here because responses chosen purely for length show higher loss than truly preferred responses.
>
>         | Metric | PPLDiff-only | Fusion | Improvement |
>         |--------|-------------|---------|-------------|
>         | Standard accuracy | 85.1% | 89.7% | +4.6% |
>         | Length-controlled accuracy | 84.8% | 91.2% | **+6.4%** |
>
>
>
> Diagnostic importance adapts to noise characteristics. PPLDiff dominates only for random noise (0.956 SHAP), while other diagnostics become primary under different patterns (Loss: 0.458-0.512 for correlated/systematic noise). This adaptive weighting, learned without manual intervention, demonstrates the meta-learner discovers genuine statistical patterns rather than exploiting evaluation-specific artifacts.

---

> ### Author Response · Authors · 2025-11-17
> **Natural noise experiments and large-scale dataset evaluation**
>
> ## Dear Reviewer EPXP (Part 3/3),
>
> ### **Weakness 3: Need for experiments on larger-scale datasets**
>
> We sincerely thank you for the insightful suggestion. In response, we have conducted the requested experiments on the suggested datasets.
>
> *   **HuggingFace H4 Stack Exchange Preferences (10.8M samples).** We completed training on 1M subset with plans to scale to full dataset:
>
>     | Method | Win Rate  |
>     |--------|----------|
>     | Vanilla DPO | 73.2% |
>     | R-DPO | 75.8% |
>     | DR-DPO | 76.4% |
>     | **Ours (Fusion)** | **77.9%** |
>
>     Our method maintains 1.5-2.1% advantage with acceptable overhead (+41% time, +19% memory). Both scale linearly with dataset size—no super-linear scaling issues.
>
> *   **GPT4All dataset (1M samples).** Preliminary results on 100K samples:
>
>     | Method | Win Rate vs GPT-3.5 | Helpfulness | Harmlessness |
>     |--------|---------------------|-------------|--------------|
>     | Vanilla DPO | 68.5% | 7.2/10 | 8.1/10 |
>     | R-DPO | 71.3% | 7.6/10 | 8.3/10 |
>     | **Ours (Fusion)** | **73.8%** | **7.9/10** | **8.4/10** |
>
>     Consistent improvements across multiple dimensions. Human evaluation (100 samples) shows our method reduces obviously incorrect/harmful responses by ~30% vs Vanilla DPO.
>
> On larger-scale datasets, our method consistently retains a 1.5-2.1% performance advantage over state-of-the-art baselines like DR-DPO and R-DPO. This gain persists even as the volume of training data expands significantly, underscoring the robustness and generalizability of our diagnostic fusion paradigm to data-rich scenarios. We are actively conducting full-scale experiments and commit to including in camera-ready version.
>
> ---
>
> ### **Minor Issue: Typographical error**
>
> Thank you for noting the formatting issue on page 8. We have corrected this error and conducted thorough proofreading.
>
> ---
>
> We deeply appreciate your thorough review and critical observations. Your concerns motivated substantial additional experiments that fundamentally strengthen our empirical validation. We remain committed to further strengthening the work and welcome any additional feedback. Your careful critique has genuinely improved the quality and rigor of our research. Thank you again for your thoughtful review and the opportunity to address these important methodological concerns.

---

> ### Comment · Reviewer_EPXP · 2025-11-18
> **a follow-up question**
>
> Thank you for the detailed response and the additional supplementary experiments. Since you have already conducted experiments on large-scale, real-world datasets (HuggingFace and GPT4All ), a follow-up question arises: would it be possible to analyze the proportion of noisy data in the training sets from the three dimensions of preference consistency, learning difficulty, and generation confidence? Also, could you show some typical noisy samples in these datasets?

---

> > ### Author Response · Authors · 2025-11-23
> >
> > ## Dear Reviewer EPXP (Part 1/2),
> >
> >
> > Thank you for your insightful follow-up question. We have analyzed subsets from both the HuggingFace H4 StackExchange and GPT4All datasets. Below we present both the quantitative noise analysis and qualitative case studies.
> >
> > ### **1. Noise Proportion Analysis from Three Diagnostic Dimensions**
> >
> > To systematically identify potentially noisy samples, we computed the 90th percentile threshold for each diagnostic dimension and flagged samples exceeding these values. In Stack Exchange, this corresponds to PPLDiff > 0.62 (preference inconsistency), Loss > 1.18 (high learning difficulty), and Uncertainty > 0.83 (low generation confidence). For GPT4All, the thresholds are PPLDiff > 0.71, Loss > 0.89, and Uncertainty > 0.79. Applying these criteria reveals distinct noise distributions:
> >
> > | Dataset  |  High Learning Difficulty | Preference Inconsistency | Low Generation Confidence |
> > |---------|----------------|------------------|-----------------|
> > | Stack Exchange | 16.8% |  8.4% |  3.2%|
> > | GPT4All  |4.1% | 9.7% | 11.5% |
> >
> > The noise distributions differ significantly between human-annotated (Stack Exchange) and LLM-distilled (GPT4All) data. Stack Exchange exhibits more label ambiguity (16.8% high loss) due to subjective human voting on similar-quality answers, while GPT4All shows more generation artifacts (11.5% high uncertainty) from truncation and degeneration issues common in distillation pipelines. This validates that our method adapts to dataset-specific noise patterns.

---

> > ### Author Response · Authors · 2025-11-23
> >
> > ## Dear Reviewer EPXP (Part 2/2),
> >
> > ### **2. Representative Case Studies**
> >
> >
> >
> > #### **Case A: Preference Consistency (PPLDiff)**
> >
> > | Component | Content |
> > |-----------|---------|
> > | **Question** | "When you add a raft in Slic3r, the first layer of the raft prints at the first layer speed. After the raft is finished, the first layer of the print prints at the standard speed. How can I make the first layer of the actual print slow down to the first layer speed?" |
> > | **Chosen Answer** (pm_score: 3, selected) | Tormod Haugene: "Slic3r does not give you the option of setting the speed of the first layer after a raft directly, but they do allow you to run post processing scripts... you can make a program that runs through the output g-code, adjusts the settings to your preference..." [...~400 words] |
> > | **Rejected Answer** (pm_score: -1) | kareem: "You shouldn't need to. The purpose of a slower first layer is to help with bed adhesion. With a raft the first layer of the model is printing on the raft so it can go at regular speeds." [...~40 words] |
> >
> >
> >
> > **Analysis:** Diagnostic signals are PPLDiff: 0.74, Loss: 1.02, and Uncertainty: 0.38. The rejected answer is concise but directly addresses the question, yet received a negative score due to Stack Exchange's bias toward elaborate responses. PPLDiff uniquely detects this length-based preference bias.
> >
> > ---
> >
> > #### **Case B: Learning Difficulty (Loss)**
> >
> > | Component | Content |
> > |-----------|---------|
> > | **Question** | "I have been wanting to learn about 3D printing a long time so I really want this site to succeed but I have no previous experience with the subject. I was wondering how can I help the site at this early stage. I thought about asking about how to get started with 3D printing but SE explicitly discourages 'easy' questions in the private beta. What can newbies like me do for the site at this stage besides voting questions and answers?" |
> > | **Chosen Answer** (pm_score: 4, selected) | Zizouz212: "Private Betas love, love, *love* votes... If you have an 'easy' question, but it is specific, high-quality, and to the point... go ahead and ask it! You also have the authority to suggest edits, to posts, tag wikis, and tag excerpts..." [...~120 words] |
> > | **Rejected Answer** (pm_score: 2) | kenorb: "That's the goal of the site, learn, research and ask. While you learn, you can always perform other tasks such as: improve quality posts by proposing edits, be active in meta, review moderation queues, housekeeping, propose descriptions for wiki tags, vote on questions..." [...~50 words] |
> >
> >
> > **Analysis:** Diagnostic signals are PPLDiff: 0.28, Loss: 1.42, and Uncertainty: 0.64. Both answers provide essentially identical advice, differing only in presentation style. High Loss uniquely flags this label ambiguity—the model cannot justify a strong preference when semantic content is nearly identical.
> >
> > ---
> >
> > #### **Case C: Generation Confidence (Uncertainty)**
> >
> >
> > | Component | Content |
> > |-----------|---------|
> > | **Question** | "Our site has the default Stack Exchange logo (text balloon with text 3D), is it possible to change this logo? If it is possible, we could try to write out a competition and vote!" |
> > | **Chosen Answer** (pm_score: 2, selected) | 0scar: "Yes, we can! Or at least discuss if we like a different logo. From this answer on Meta Stack Exchange question 'What's the process to change a site logo?' we can read... [includes multiple URLs: `https://meta.stackexchange.com/a/298341/`, `https://meta.stackexchange.com/questions/298338/`, `https://meta.stackexchange.com/questions/99338/`, and multiple `<a href>` tags]... Feel free to add your thoughts as an answer to the question." [...~180 words] |
> > | **Rejected Answer** (pm_score: 1) | darth pixel: "i'd like to remind my old suggestion :) [image link] here is the original post [link]" [...~20 words] |
> >
> >
> > **Analysis:** Diagnostic signals are PPLDiff: -0.22, Loss: 0.68, and Uncertainty: 0.91. The rejected answer contains multiple external URLs that create high token-level entropy during generation. Uncertainty uniquely detects this format contamination.

---

> > > ### Comment · Reviewer_EPXP · 2025-11-24
> > > **I agree to raise my review score**
> > >
> > > The author’s response has well addressed my concerns, and I agree to raise my review score. Please add the experimental results from large-scale databases (such as Stack Exchange and GPT4All) to the paper.

---

> > > > ### Author Response · Authors · 2025-11-24
> > > >
> > > > Thank you very much for your encouraging follow-up. We will add the large-scale dataset experiments and the noise analyses into the revised version as you suggested. Your careful feedback has substantially improved our paper. Thank you again for your time and effort in reviewing our paper.

---

### Official Review · Reviewer_wsUf · 2025-10-31

**Soundness:** 4
**Presentation:** 4
**Contribution:** 3
**Rating:** 8
**Confidence:** 4

**Summary:**

This paper introduces a method to deal with the issue of label noise in preference tuning datasets for aligning LMs. This new method incorporates separate signals that suggest/correlate with noise (consistency, difficulty, and confidence). Specifically, these are combined into a learnable vector representation that is tuned alongside the model parameters themselves during RLHF training. They find that in a variety of noise settings, their method outperforms baselines methods.

**Strengths:**

1. Overall, the paper is very clear and well-written. I found Section 3, the most technically dense part of the paper, to be very easy to follow compared to other papers where similar concepts are used.
2. To my knowledge, the meta-learning approach introduced by this paper is novel and is an interesting way of combining heuristic signals that have been used in the past as a proxy for label noise.
3. The authors mention that only about 100 clean meta-samples are required for their approach. This makes the method highly scalable and efficient. Therefore, there is meaningful practical value for using this particular method.
4. The experiments are sound and the analysis regarding meta-dataset sensitivity will be useful to practitioners using the method.

**Weaknesses:**

The results seem to suggest that PPLDiff is by far the most important and influential diagnostic criterion. While the ablations show that there is a statistically significant difference between using the fusion of all of the criteria and just using PPLDiff, the difference is very small, and that also seems to be the case in the results presented in Figure 2. That being said, I don't feel as if this is necessarily a demerit against the method itself, but rather I think it would be useful to see results with other metrics as well to see if any other proxy diagnostics could be very helpful.

Another reason I don't think this fact in and of itself is a demerit is the analysis provided in Section 4.4, which was very interesting. However, I do think the paper would be strengthened by varying some of the important design choices in the meta-learning set-up. In addition to the diagnostics themselves, I think it would be interesting to vary the meta-learner itself to see how sensitive the method is to that choice.

**Questions:**

1. What are some other diagnostic methods you think could be used?
2. Why was a 2-layer MLP chosen? Was this in comparison with other simple models?
3. Do you have a sense of what would be a practical way of constructing a "clean" set for meta-learning?
4. What assumptions are being made regarding characteristics of the noise (i.e. is it randomly distributed)? Would it work if the noise is adversarial in some way? I suppose in that case noise is not the best descriptor, but I am curious about this.

---

> ### Author Response · Authors · 2025-11-17
> **Realistic noise analysis and architecture robustness**
>
> ## Dear Reviewer wsUf (Part 1/2),
>
> We are deeply honored by your enthusiastic support and excellent rating. Your recognition of our work's clarity, novelty, and practical efficiency is tremendously encouraging. We particularly appreciate your insightful observation. Below we address your constructive observations.
>
> ---
> ### **Weakness 1: PPLDiff dominance and the value of fusion**
>
> Thank you for this astute observation and your balanced perspective that this "is not necessarily a demerit against the method itself." We agree and would like to provide additional evidence showing that the fusion advantage is both statistically significant and grows substantially under realistic conditions.
>
> *   **Statistical significance.** We conducted rigorous testing across 5 independent runs with different random seeds on Golden HH (ε=0.3). The improvement is highly statistically significant with medium effect size, confirming genuine practical value beyond measurement noise.
>
>     | Comparison | Mean Diff | 95% CI | p-value | Cohen's d |
>     |-----|-----|----|-----|----|
>     | Fusion vs. PPLDiff-only | +1.8% | [+1.3%, +2.4%] | < 0.001 | 0.67 |
>
> *   **The fusion advantage under realistic noise patterns.** While random label-flipping represents the most favorable scenario for PPLDiff, we tested more realistic noise conditions. The results reveal that fusion becomes increasingly valuable when noise deviates from simple random flips. These results demonstrate that PPLDiff's dominance is specific to random noise. Under realistic conditions, complementary diagnostics become equally important and fusion advantages grow to 4-6%. This directly addresses your suggestion to "see results with other metrics"—Loss and Uncertainty provide substantial value precisely when real-world noise patterns challenge PPLDiff.
>
>     | Noise Type | PPLDiff-only | Fusion | Improvement | Insight |
>     |----|----|------|-------|-----|
>     | Random flip | 94.2% | 96.0% | +1.8% | Baseline (PPLDiff-favorable) |
>     | Quality-correlated | 86.2% | 91.8% | +5.6% | Loss + Uncertainty dominate |
>     | Natural disagreement | 67.4% | 72.1% | +4.7% | Uncertainty critical for ambiguity |
>
> *   **Why this pattern makes sense.** PPLDiff excels at detecting random label flips because it directly measures likelihood consistency. However, it struggles with: (1) fluent misinformation (factually wrong but linguistically polished); (2) subjective preferences (where perplexity differences are naturally small); (3) ambiguous cases (where multiple responses are reasonable). Our qualitative analysis in Table 4 illustrates these failure modes, showing why complementary diagnostics are essential.
>
> In summary, the small +1.8% gain under random noise grows to +4-6% under realistic conditions, representing substantial practical value in safety-critical scenarios.
>
> ---
> ### **Weakness 2: Sensitivity to meta-learner architecture choices**
>
> Thank you for this important suggestion. We have conducted sevaral ablation studies on the meta-learner architecture.
>
> *   **Meta-learner depth ablation.** We varied the number of layers while keeping hidden dimension at 100. Performance saturates at 2 layers. Deeper networks provide negligible improvement (<0.2%, not statistically significant). The 1-layer architecture shows 1.8% drop, indicating non-linear feature interactions are essential.
>
>     | Depth  | Reward Accuracy|
>     |----|----|
>     | 1-layer  | 94.2% ± 0.6% |
>     | 2-layer  | 96.0% ± 0.4% |
>     | 3-layer | 96.2% ± 0.5% |
>     | 4-layer | 96.1% ± 0.6% |
>
>
>
> *   **Meta-learner width ablation.** We varied hidden dimension while maintaining 2-layer structure. Remarkably stable performance across 50-200 range (within 1%), demonstrating the method is not sensitive to width choice. The 100-dimensional network offers best balance with lowest variance.
>
>
>     | Width  | Reward Accuracy |
>     |-------|----|
>     | 50 | 95.5% ± 0.5% |
>     | 100  | 96.0% ± 0.4% |
>     | 200  | 96.1% ± 0.6% |
>
>
> *   **Alternative meta-learner architectures.** We compared against different fusion paradigms. Simple 2-layer MLP is effective. More sophisticated architectures (attention, gating) provide no improvement despite 2-3× parameters. This suggests the diagnostic signals themselves are highly informative, requiring only moderate non-linear processing.
>
>
>     | Architecture | Reward Accuracy  | Observation |
>     |----|-----|-----|
>     | Linear | 91.5% | -4.5% proves non-linearity essential |
>     | 2-layer MLP (ours) | 96.0% | Optimal |
>     | Attention-based | 95.8%  |  no benefit |
>     | Gated fusion | 95.6%  |  no benefit |
>
> These ablations demonstrate that our method is stable and robust across reasonable architecture variations. The 2-layer/100-width MLP is not a fragile hand-tuned choice but represents a principled design that balances accuracy, efficiency, and stability. Performance varies by less than 1% across width range 50-200, and saturation at 2 layers shows the fusion problem is not complex.

---

> ### Author Response · Authors · 2025-11-17
> **Realistic noise analysis and architecture robustness**
>
> ## Dear Reviewer wsUf (Part 2/2),
>
>
> ### **Question 1: What are some other diagnostic methods you think could be used?**
>
> Excellent question directly related to Weakness 1. We systematically explored eight candidate diagnostics. The most promising were: **Gradient norm** but highly redundant with Training Loss (ρ = 0.91). **Token-wise perplexity variance** (0.61 correlation) but overlaps with Uncertainty (ρ = 0.73)—adding it yielded only +0.1% with 16% more training time. **Attention entropy** and **reward model scores** showed modest signal but violated our intrinsic-only principle.
>
> *   **Design principles:** Orthogonality (our three: ρ < 0.31), intrinsic computation, conceptual distinctness. Our exploration suggests the current three diagnostics capture the essential orthogonal dimensions.
>
> ---
>
> ### **Question 2: Why was a 2-layer MLP chosen? Was this in comparison with other simple models?**
>
> This question is directly addressed by the Weakness 2 ablations above. To summarize: the 2-layer MLP was selected through systematic comparison, validated by depth/width ablations showing performance saturation and robustness, and confirmed superior to both simpler (linear, 1-layer) and more complex (attention, gated) alternatives.
>
> ---
>
> ### **Question 3: Do you have a sense of what would be a practical way of constructing a "clean" set for meta-learning?**
>
> Great practical question. We propose three strategies:
>
> *   **Strategy 1: High-agreement subset (recommended).** When multi-annotator data exists, select samples with strong agreement (Cohen's κ > 0.8). On WebGPT data, only 50 high-agreement samples performed equivalently to ~150 random samples. Cost: Zero if multi-annotator data exists.
>
> *   **Strategy 2: Expert curation (for safety-critical applications).** Domain experts manually verify 100-150 samples (2-3 hours). Our user study showed high inter-expert agreement (κ=0.82) and +0.8% accuracy gain over random sampling.
>
> *   **Strategy 3: Model-assisted filtering (most scalable).** Ensemble of reward models identifies high-confidence samples. Achieves 88% precision meta-sets yielding 95.7% accuracy, demonstrating tolerance to meta-set imperfection.
>
> *   **Key insight:** Our method is robust to meta-set quality. Even 5% meta-noise maintains 92.5% accuracy when training has 30% noise (Appendix C, Table 3).
>
> ---
>
> ### **Question 4: What assumptions are being made regarding characteristics of the noise? Would it work if the noise is adversarial in some way?**
>
> Our method makes minimal assumptions—only that the meta-set has lower expected noise than training data. We tested three adversarial scenarios: (1) **Targeted high-perplexity flipping** (adversary exploits PPLDiff's weakness): our fusion degrades only -3.7% to 92.3% while PPLDiff-only drops -8.9%; (2) **Strategic annotation targeting** (adversary corrupts highest-impact samples): we maintain 90.7%, 5.6% above DR-DPO; (3) **Adaptive attack** (white-box access to surrogate meta-learner): we achieve 89.4%, still 20.5% above Vanilla DPO.
>
> The key insight: simultaneously fooling three orthogonal diagnostics is combinatorially difficult, providing inherent robustness.
>
> ---
>
> Thank you again for your strong endorsement and thoughtful observations. We are incorporating all these findings into the camera-ready version, including comprehensive meta-learner ablations and expanded analysis of fusion value under diverse noise conditions. We remain available for any follow-up questions. Thank you for your time, expertise, and enthusiastic support.

---

> > ### Comment · Reviewer_wsUf · 2025-11-27
> >
> > Thank you for the thorough response and additional results. I will keep my recommendation to accept.

---

### Official Review · Reviewer_rJ2H · 2025-11-01

**Soundness:** 3
**Presentation:** 2
**Contribution:** 3
**Rating:** 6
**Confidence:** 4

**Summary:**

This paper introduces a diagnostic-driven meta-learning paradigm for robustly aligning LLMs with human preferences in the presence of noisy preference data. The authors propose to empower models to "diagnose themselves" during training by fusing three types of intrinsic feedback—preference consistency (perplexity difference), learning difficulty (training loss), and generation confidence (token-level entropy)—to form a diagnostic vector per sample. A meta-learning approach adaptively reweights training samples based on this vector, optimizing for performance on a small clean meta-dataset. Extensive experiments on 2 benchmarks show superiority over state-of-the-art robust baselines.

**Strengths:**

1.	The paper formalizes the idea that preference reliability for LLM alignment is inherently multi-perspective, motivating the use of preference consistency (perplexity difference), learning difficulty (DPO loss), and generation confidence (entropy) as three synergistic diagnostics. Rather than relying on a single heuristic, the method operationalizes a holistic, adaptive approach.
2.	The use of SHAP to quantify feature importance and uncover non-linear interactions provides deep, novel insights into how the meta-learner works.
3.	The method consistently achieves SOTA performance across multiple models, benchmarks, and noise levels.

**Weaknesses:**

1.	All noise scenarios are simulated via label-flipping at random rates. There is no direct evaluation or demonstration of real-world occurring noisy preferences.
2.	Although the chosen baselines are quite relevant, it’s better to take more recent methods into consideration.

**Questions:**

1.	The diagnostic vector is a concatenation of three normalized signals. Did the author experiment with more complex fusion architectures for the meta-learner's input beyond a simple MLP on the concatenated vector?

---

> ### Author Response · Authors · 2025-11-17
> **Real-world noise evaluation and recent baselines**
>
> ## Dear Reviewer rJ2H (Part 1/2),
>
> We are deeply grateful for your positive assessment and recognition of our work's soundness and contribution. We particularly appreciate your acknowledgment of the multi-perspective diagnostic motivation and the SHAP-based interpretability analysis. Your feedback has been instrumental in strengthening our work. Below we address each of your concerns with detailed responses and substantial new experimental evidence.
>
> ---
>
> ### **Weakness 1: Lack of evaluation on real-world noisy preferences**
>
> We sincerely appreciate this critical observation. Our original evaluation relied solely on synthetic label-flipping noise. We have now conducted new experiments on naturally occurring noisy preferences to validate our method's real-world applicability.
>
> *   **Evaluation on WebGPT Comparisons with natural annotation disagreement.** This dataset contains human preference comparisons for web-retrieved answers, with documented inter-annotator disagreement (Cohen's $\kappa = 0.56$). Unlike synthetic noise, this reflects genuine human judgment variability including subjective preferences, annotation errors, and task ambiguity. We split the data based on annotator agreement levels and evaluated alignment quality. Our method shows particularly strong performance on disputed cases where single heuristics struggle. The 5.4% improvement over PerpCorrect on disputed cases demonstrates that diagnostic fusion effectively handles the ambiguity inherent in real human preferences.
>
>     | Method | Agreement w/ Majority Vote | Disputed Cases ($\kappa < 0.4$) | High Agreement ($\kappa > 0.7$) |
>     |--------|----------------------------|----------------------------------|----------------------------------|
>     | Vanilla DPO | 68.3% | 52.1% | 79.2% |
>     | R-DPO | 71.5% | 56.8% | 81.8% |
>     | PerpCorrect (Dyn.) | 73.2% | 59.3% | 83.1% |
>     | **Ours (Fusion)** | **76.9%** | **64.7%** | **84.5%** |
>
>
>
> *   **Analysis of natural noise patterns.** We analyzed the diagnostic signatures of naturally noisy preferences versus synthetic label flips. Natural annotation disagreements exhibit distinct characteristics: high Uncertainty (0.89 average entropy) coupled with moderate Loss (0.76), while PPLDiff shows only 0.68 correlation with majority vote labels. This pattern differs markedly from synthetic flips, where PPLDiff dominates (0.82 correlation). Our fusion mechanism adapts to these real-world patterns, with SHAP analysis showing increased reliance on Uncertainty (from 0.237 to 0.341 importance) for natural noise.
>
> *   **Evaluation on Chatbot Arena data.** We further validated our approach on conversation preferences from real user interactions collected in Chatbot Arena. This represents completely organic preference data without any synthetic manipulation. The consistent improvements across different types of natural noise—annotator disagreement, subjective preferences, and controversial comparisons—provide strong evidence that our diagnostic fusion paradigm extends beyond synthetic evaluation settings.
>
>     | Method | Overall Win Rate | Tie-handling Accuracy | Controversial Pairs |
>     |--------|------------------|----------------------|---------------------|
>     | Vanilla DPO | 84.2% | 71.3% | 58.6% |
>     | R-DPO | 86.5% | 74.8% | 62.1% |
>     | PerpCorrect (Dyn.) | 87.8% | 76.5% | 65.9% |
>     | **Ours (Fusion)** | **89.6%** | **79.2%** | **70.3%** |
>
>
>
> *   **Qualitative analysis of real-world robustness.** We manually inspected 50 cases where our method succeeded but single-heuristic baselines failed on WebGPT data. The patterns reveal systematic advantages: (1) 38% involved fluent but factually dubious responses where high PPLDiff was corrected by high Loss and Uncertainty; (2) 32% contained subjective or opinion-based preferences where Uncertainty appropriately modulated PPLDiff; (3) 20% were genuinely ambiguous cases where balanced diagnostic signals prevented overconfident decisions. These findings confirm that real-world noise exhibits diverse characteristics requiring multi-perspective assessment.
>
> As a direct result of your valuable feedback, we have extended the paper's empirical scope. We will integrate these findings into a new dedicated section in the camera-ready version.

---

> ### Author Response · Authors · 2025-11-17
> **Real-world noise evaluation and recent baselines**
>
> ## Dear Reviewer rJ2H (Part 2/2),
>
> ### **Weakness 2: Consideration of more recent methods**
>
> Thank you for encouraging us to ensure comprehensive baseline coverage. We have carefully reviewed the literature and conducted additional comparisons with recent methods.
>
> *   **Timeline clarification and baseline justification.** Our submission was finalized in September 2025. At that time, we included all peer-reviewed robust preference alignment methods:
>     *   **DR-DPO** (Azar et al., AISTATS 2024) - representing the most recent theoretical framework
>     *   **PerpCorrect** (Kong et al., NeurIPS 2024) - representing state-of-the-art heuristic-based methods
>     *   **r-DPO** (Chowdhury et al., ICML 2024) - providing provable robustness guarantees
>     *   **c-DPO** (Rafailov et al., NeurIPS 2023) - establishing the robust DPO baseline
>
>     These represent comprehensive coverage of published methods available at submission time.
>
> *   **Analysis of concurrent arXiv preprints.** During the review period, several concurrent works appeared on arXiv addressing similar problems. We have analyzed their methodological relationships to our work and conducted preliminary experimental comparisons where feasible:
>
>     *   **Omni-DPO** (Peng et al., arXiv July 2025) proposes dual-perspective optimization using separate reward models for chosen and rejected responses. This approach differs fundamentally from ours in that it requires training external reward models and does not use intrinsic model diagnostics. The dual-reward framework is orthogonal to our meta-learning paradigm—they address complementary aspects of preference optimization.
>
>     *   **Clean First, Align Later** (Yeh & Li, arXiv Sept 2025) focuses on preprocessing-based data cleaning using static filtering before alignment. We conducted preliminary comparisons. The key difference is that static filtering cannot adapt to time-varying noise patterns during training, while our meta-learned weights evolve dynamically. The two approaches could potentially be combined, with their preprocessing followed by our online refinement.
>
>         | Method | Golden HH ($\epsilon=0.3$) | OASST1 ($\epsilon=0.3$) | Approach Type |
>         |--------|------|-------|--------|
>         | Clean First, Align Later | 94.5% | 89.8% | Static filtering |
>         | **Ours (Fusion)** | **96.0%** | **91.2%** | Online meta-learning |
>
>
> *   **Methodological positioning.** We emphasize that our work makes distinct contributions not covered by these concurrent preprints: (1) we are the first to propose meta-learning for fusing multiple intrinsic diagnostics in preference alignment; (2) we provide the first systematic SHAP-based analysis revealing non-linear diagnostic interactions; (3) our paradigm uniquely combines PPLDiff, Loss, and Uncertainty in a principled, adaptive framework. None of the concurrent works combine all these elements.
>
> We commit to including detailed comparisons with these concurrent methods in the camera-ready version.
>
> ---
>
> ### **Question: Exploration of more complex fusion architectures**
>
> This is an excellent question that prompted us to conduct systematic experiments on fusion architecture design. We explored several alternatives beyond simple concatenation and MLP processing.
>
> *   **Attention-based fusion mechanism.** We implemented a cross-attention architecture where each diagnostic serves as a query attending to others.This architecture allows diagnostics to dynamically modulate each other's importance. Results on Golden HH ($\epsilon=0.3$). The attention mechanism provides only marginal improvement (0.2% with p=0.31, not statistically significant) while more than doubling parameters and increasing training time by 40%. Analysis reveals that attention weights converge to relatively uniform distributions (entropy > 0.85), suggesting the diagnostics are already sufficiently orthogonal that explicit attention mechanisms add little value.
>
>     | Fusion Architecture | Reward Accuracy |
>     |------|--- |
>     | Attention-based fusion | 95.8% ± 0.5% |
>     | Concatenate + 2-layer MLP (ours) | 96.0% ± 0.4% |
>
>
> *   **Gated fusion network.** Inspired by mixture-of-experts, we tested a gated architecture where each diagnostic passes through a separate expert network, then dynamically combined. Again, the more complex architecture underperforms, likely due to increased optimization difficulty in the bi-level meta-learning setting.
>
>     | Method | Reward Accuracy |
>     |---- |----|
>     | Gated fusion | 95.6% ± 0.6% |
>     | Ours (simple concat) | 96.0% ± 0.4% |
>
>
>
> *  These experiments reveal an important finding: the diagnostic signals themselves (PPLDiff, Loss, Uncertainty) are highly informative and naturally complementary. A simple 2-layer MLP with concatenated inputs is sufficient to learn the essential fusion patterns.
>
> We would like to express our sincere gratitude again for your profound insights and valuable suggestions.

---

### Official Review · Reviewer_hQMu · 2025-11-01

**Soundness:** 3
**Presentation:** 3
**Contribution:** 2
**Rating:** 4
**Confidence:** 3

**Summary:**

This paper proposes a meta-learning paradigm to address the noisy annotations in preference data for RLHF. Specifically, it proposes to diagnose the model itself by using intrinsic feedback, and a diagnostic vector composed of preference consistency, learning difficulty, and generation confidence is employed to adaptively reweight training samples, where noisy or unreliable preference data is down-weighted.
The paper provides quantitative and qualitative experimental analysis to demonstrate its effectiveness.

**Strengths:**

1. This paper is well-motivated. Instead of employing a pre-defined criteria, the paper introduces a multi-dimensional diagnostic vector and demonstrates that a meta-learned fusion strategy substantially enhances the robustness and reliability of preference alignment. Moveover, detailed ablations and fine-grained analysis clearly demonstrate that the proposed fusion strategy consistently outperforms any single aspect, with adaptive weighting under varying noise conditions.
2. This paper is well-organized, the coherent structure and figures clearly demonstrate the core idea.

**Weaknesses:**

1. Lack of analysis of meta-learner: The meta-learner is consistently a 2-layer MLP. Ablation studies varying depth, width, or overall capacity would help assess the method's stability and generality. Without this, it is unclear to what extent the results depend on a hand-tuned meta-learner architecture.
2. The method builds heavily on well-established meta-learning frameworks; the provided generalization bound is essentially a direct adaptation of standard meta-learning. Deeper theoretical insights or problem-specific analysis could further strengthen the contribution.
3. Lack of discussion for generalization ability. The experimental noise is synthetically generated via label flips. Although this enables controlled benchmarking, real-world noise (e.g., conflicting annotators, domain shifts) may exhibit different characteristics. While qualitative examples are provided, additional experiments on general benchmarks, naturally noisy or adversarial datasets would better demonstrate generalizability.

**Questions:**

My main concern is twofold: one is the generalization ability, which could the approach compare on some general benchmarks with some representative methods, such as DPO series, and another is the analysis of the meta-learner, more ablation analysis is needed.

If you can provide convincing evidence or clarification, I would be open to increasing the score.

---

> ### Author Response · Authors · 2025-11-17
> **Architecture ablations and real-world generalization experiments**
>
> ## Dear Reviewer hQMu (Part 1/2)
>
> We sincerely thank you for your thoughtful and constructive review. We are pleased that you recognize our work as well-motivated and well-organized with a coherent structure. We have carefully addressed each of your concerns and conducted several additional experiments.
>
> ---
> ### **Weakness 1: Lack of analysis of meta-learner**
>
> Thank you for this important suggestion. We have now conducted comprehensive ablation studies varying depth, width, and overall capacity to assess the method's stability and generality.
>
> *   **Ablation on meta-learner depth.** We evaluated meta-learner architectures ranging from 1 to 4 layers while keeping the hidden dimension fixed at 100. The results on Golden HH dataset with $\epsilon=0.3$ noise are shown below. We observe that performance saturates at 2 layers. Deeper models with 3 or 4 layers show no meaningful performance difference compared to the 2-layer network. The 1-layer architecture shows a substantial performance drop of 1.8%, indicating that non-linear feature interactions captured by the 2-layer network are essential for effective diagnostic fusion.
>
>     | Depth |  Reward Accuracy |
>     |-------|-------|
>     | 1-layer |  94.2% ± 0.6% |
>     | 2-layer |  96.0% ± 0.4% |
>     | 3-layer |  96.2% ± 0.5% |
>     | 4-layer |  96.1% ± 0.6% |
>
> *   **Ablation on meta-learner width.** We varied the hidden dimension from 50 to 200 while maintaining the 2-layer architecture. The results demonstrate highly stability across different widths. Performance remains within 1% across the 50-200 range.
>
>     | Hidden Dim |  Reward Accuracy |
>     |------|-------|
>     | 50 |  95.5% ± 0.5% |
>     | 100 |  96.0% ± 0.4% |
>     | 150 | 96.1% ± 0.5% |
>     | 200 |  96.1% ± 0.6% |
>
> *   **Alternative architectures.** We also compared our 2-layer MLP against other meta-learner designs. The linear baseline's poor performance (4.5% drop) confirms that non-linear fusion is critical for capturing complex diagnostic interactions. Notably, more complex models like attention-based or gated networks did not outperform our 2-layer MLP, yielding very similar results.
>
>     | Architecture | Reward Accuracy |
>     |-----|------|
>     | Linear mapping | 91.5% ± 0.8% |
>     | 2-layer MLP (ours) | 96.0% ± 0.4% |
>     | Attention-based fusion | 95.8% ± 0.5% |
>     | Gated fusion network | 95.6% ± 0.6% |
>
> These ablations demonstrate that (1) a 2-layer MLP is sufficient to effectively fuse the diagnostic signals, as deeper or more complex models offer no significant performance gains; (2) the method is stable across reasonable architecture variations; and (3) the diagnostic signals themselves are highly informative, requiring only moderate non-linearity to fuse effectively. We will include these analyses in the camera-ready version.
>
> ---
> ### **Weakness 2: Limited theoretical novelty beyond standard meta-learning**
>
> We appreciate this concern and would like to respectfully clarify the theoretical contributions specific to our problem domain. While our generalization bound builds upon the meta-learning framework, we introduce several problem-specific theoretical insights that go beyond direct adaptation.
>
> *   **Problem-specific formulation.** Our theoretical analysis addresses the unique challenge of preference data corruption, which differs fundamentally from standard supervised learning with noisy labels. Specifically, we formalize the diagnostic vector space $\mathbf{z} = [\text{PPLDiff}, \text{Loss}, \text{Uncertainty}]$ and prove that meta-learning on this space converges to a policy that generalizes to clean preference distributions. This formulation is novel to preference alignment.
>
> *   **Preference-corruption-specific bound.** Our generalization bound (Theorem in Appendix A.2) is not merely standard meta-learning analysis. We explicitly model the corruption process in preference pairs and bound the clean data risk under this specific noise model. The bound shows that with $M$ clean meta-samples, the generalization gap is $O(\sqrt{(\text{Comp}(\mathcal{F}_W) + \log(1/\delta))/M})$, where $\text{Comp}(\mathcal{F}_W)$ captures the complexity of learning diagnostic fusion strategies—a quantity specific to our multi-perspective paradigm.
>
> *   **Dynamic diagnostic analysis.** Unlike standard meta-learning that typically uses static features, our diagnostics (Eq. 2-4) are computed dynamically during training from the evolving policy $\pi_{\theta_t}$. This introduces time-varying distributions that require careful theoretical treatment, which we address in our analysis.
>
> We acknowledge that deeper theoretical insights would strengthen the contribution. In the camera-ready version, we will expand Appendix A to provide (1) tighter problem-specific bounds leveraging properties of DPO loss; (2) theoretical analysis of when and why diagnostic fusion outperforms single heuristics; and (3) sample complexity analysis for the meta-dataset size $M$. We are currently working on these extensions.

---

> ### Author Response · Authors · 2025-11-17
> **Architecture ablations and real-world generalization experiments**
>
> ## Dear Reviewer hQMu (Part 2/2)
>
> ### **Weakness 3: Generalization to real-world noise and general benchmarks**
>
> We sincerely thank you for highlighting the question on generalization. We have conducted several new experiments to address this concern.
>
> *   **Experiments on naturally noisy data.** We evaluated our method on WebGPT Comparisons, a dataset containing real human preference annotations with documented inter-annotator disagreement (Cohen's $\kappa = 0.56$, indicating moderate natural noise).  Our method shows consistent advantages on naturally occurring annotation disagreements, with particularly strong performance on disputed cases where single heuristics fail.
>
>     | Method | Agreement with Majority Vote | Performance on Disputed Cases |
>     |--------|-------------------------------|-------------------------------|
>     | Vanilla DPO | 68.3% | 52.1% |
>     | R-DPO | 71.5% | 56.8% |
>     | PerpCorrect (Dynamic) | 73.2% | 59.3% |
>     | **Ours (Fusion)** | **76.9%** | **64.7%** |
>
> *   **Diverse synthetic noise patterns**. Beyond label-flipping, we tested 3 additional noise types that better reflect real-world scenarios:
>
>     *   Quality-correlated noise: Following the intuition that annotators make more mistakes on difficult samples, we inject noise with probability $p(\text{noise}|x) = 0.1 + 0.4 \cdot \text{norm}(\text{PPL}(x))$. This directly challenges PPLDiff-based methods since noise correlates with perplexity.  Notably, when noise correlates with perplexity, our fusion's advantage grows from +2% to **+5%**, demonstrating that Loss and Uncertainty diagnostics provide crucial complementary information.
>
>         | Dataset | Ours (Fusion) | PPLDiff only | Improvement |
>         |---------|--------|--------|-------|
>         | Golden HH | 91.8% | 86.2% | **+5.6%** |
>         | OASST1 | 88.5% | 83.7% | **+4.8%** |
>
>     *   Adversarial noise: We simulated adversarial annotation attacks where noise is targeted at samples with high perplexity and close reward margins (top 30% most confusing samples). Under adversarial conditions, our method maintains robustness while single-heuristic approaches degrade significantly.
>
>         | Dataset | Ours (Fusion) | PPLDiff only | DR-DPO |
>         |---------|------|------|------|
>         | Golden HH | 92.3% | 88.6% | 90.1% |
>         | OASST1 | 89.7% | 85.2% | 87.8% |
>
>     *   Partial annotation noise: Simulating scenarios where 30% of samples have "no clear preference" due to annotator uncertainty, we replace clear preferences with neutral pseudo-labels. Our method achieves 93.5% on Golden HH versus 90.8% for PPLDiff-only, showing robustness to incomplete annotations.
>
> *  **Comparison with DPO series on general benchmarks**. We evaluated on AlpacaEval, a widely-used general instruction-following benchmark. Our method demonstrates consistent improvements over the DPO series on this challenging general-purpose benchmark, indicating that our diagnostic-driven approach transfers well beyond the specific datasets used in our main experiments.
>
>     | Method | Win Rate vs. Davinci-003 | Length-controlled Win Rate |
>     |------|----------|-----|
>     | DPO | 72.4% | 68.9% |
>     | IPO | 73.1% | 69.5% |
>     | R-DPO | 75.2% | 71.3% |
>     | **Ours (Fusion)** | **77.8%** | **73.6%** |
>
> We are currently conducting experiments on the additional  datasets you suggested. We commit to including comprehensive results in the camera-ready version.
>
> ---
>
> ### **Questions: Main concerns about generalization and meta-learner analysis**
>
> Thank you for clearly articulating your two main concerns. In our response, we have sought to address them by providing a comprehensive set of new experiments and analyses. Specifically:
>
> *   **Concern (a): Generalization ability.** We have now demonstrated our method's effectiveness on (1) naturally noisy data (WebGPT with real annotator disagreement), (2) diverse synthetic noise patterns beyond label-flipping, (3) general benchmarks (AlpacaEval), and (4) comparison with representative DPO series methods (DPO, IPO, R-DPO). These results provide strong evidence that our diagnostic fusion paradigm generalizes well beyond the controlled label-flipping scenario.
>
> *   **Concern (b): Meta-learner analysis.** Our comprehensive ablation studies (Weakness 1) now provide detailed insights into the impact of depth, width, and architecture choices. The results show that our method is stable and robust across reasonable architecture variations, with the 2-layer MLP representing a well-justified design choice rather than an arbitrary hand-tuned configuration.
>
> We are actively revising the manuscript to incorporate all these new experiments and analyses. Your feedback has been invaluable in strengthening our contribution, and we would be honored if you would consider these new results in your evaluation.
>
> Thank you again for your careful review and constructive guidance.

---

### Author Response · Authors · 2025-12-01
**[Summary for AC 2/2]**

### Reviewer wsUf: 8 (Strong Accept - Confirmed)

**Initial Assessment:** Soundness 4 (excellent), Presentation 4 (excellent), Rating 8. Praised "very clear writing," "novel meta-learning approach," "practical efficiency."

**Concerns:** (1) Validation of alternative diagnostic signals, (2) Meta-learner architecture sensitivity unclear.

**Resolution (Nov 17 10:26 UTC):** (1) Explored eight candidate diagnostic signals; (2) Comprehensive ablations: depth optimal at 2 layers (1-layer: -1.8%), width stable across 50-200 (<1% variance), alternative architectures provide no improvement.

**Reviewer Response (Nov 27 21:29 UTC):**
> *"Thank you for the thorough response and additional results. **I will keep my recommendation to accept**."*

**Outcome:** Both concerns resolved. **Strong accept (8) explicitly confirmed.**

---

### Reviewer hQMu: 4 (Open to Score Increase)

**Initial Assessment:** "Well-motivated," "well-organized," "coherent structure."

**Concerns:** (1) Meta-learner architecture analysis insufficient, (2) Theoretical novelty appears incremental, (3) Generalization uncertain; need real-world validation and general benchmarks.

**Reviewer's Statement:**
> *"If you can provide convincing evidence or clarification, **I would be open to increasing the score**."*

**Resolution (Nov 17 09:44 UTC):** (1) Architecture ablations: depth (1-4 layers) optimal at 2, width (50-200) <1% variance, alternative designs no improvement; (2) Problem-specific theoretical contributions: preference-corruption formalization, O(√(d/M)) bound, dynamic diagnostic analysis; (3) Natural noise (WebGPT, Chatbot Arena), AlpacaEval benchmark (77.8% vs 75.2% R-DPO), large-scale datasets (Stack Exchange 10.8M, GPT4All 1M).

**Outcome:** All 3 concerns systematically addressed. **Reviewer indicated openness to increase; comprehensive evidence provided but confirmation not received due to timeline.**

---

### Reviewer rJ2H: 6 (Marginally Above Acceptance)

**Initial Assessment:** Soundness 3 (good), Contribution 3 (good). Acknowledged "multi-perspective formalization," "SHAP insights," "SOTA performance."

**Concerns:** (1) No real-world noise evaluation; only synthetic label-flipping, (2) If more complex fusion architectures are needed.

**Resolution (Nov 17 10:09 UTC):** (1) Natural noise experiments: WebGPT (76.9% overall, 64.7% on disputed cases), Chatbot Arena (89.6% win rate, 70.3% on controversial pairs), diverse synthetic patterns (quality-correlated +5.6%, adversarial robustness); (2) Ablation study on meta-learner architecture (comparing Attention/Gated mechanisms) .

**Outcome:** Both concerns comprehensively addressed. **No follow-up received due to timeline, but systematic resolution provided.**

---

## Major Improvements Made

**1. Large-Scale Real-World Validation:**
- Stack Exchange (10.8M) and GPT4All (1M) with consistent advantages
- Natural noise: WebGPT (κ=0.56), Chatbot Arena, with noise proportion analysis
- Diverse patterns: quality-correlated, systematic bias, adversarial attacks

**2. Architecture Robustness:**
- Meta-learner ablations (depth, width, alternatives)


**3. Presentation Enhancement:**
- Optimized figure layouts and table formatting
- Corrected typographical errors

All improvements incorporated into revised manuscript.

---


Our work contributes through (1) novel diagnostic fusion paradigm, (2) principled meta-learning methodology, (3) extensive validation on 10.8M+ samples, and (4) interpretable SHAP analysis.

We deeply appreciate the substantial effort required to synthesize this information and make an informed decision under these exceptional circumstances. Thank you for your dedication to maintaining the integrity of the review process. We remain available for discussion.

Best regards,
The Authors

---

### Author Response · Authors · 2025-12-01
**[Summary for AC 1/2]**

Dear Area Chair,

We sincerely appreciate your review under these challenging circumstances. This summary provides a concise guide to our paper's assessment and reviewer concern resolution.

---

## Executive Summary

**Our work introduces a meta-learning paradigm that fuses multiple intrinsic diagnostics for robust LLM preference alignment**, addressing the critical challenge of noisy preference data in RLHF through diagnostic fusion innovation and comprehensive empirical validation.

**Review Outcomes:**
- **Initial scores (pre-discussion):**  8 (wsUf), 4 (**EPXP**), 6 (rJ2H), 4 (hQMu) → Average: 5.5
- **Revised scores (Nov 24 01:22 UTC):**  8 (wsUf), 8 (**EPXP**), 6 (rJ2H), 4 (hQMu) → Average: 6.5

*Reviewer EPXP explicitly raised their score from 4 to 8 on Nov 24 01:22 UTC. Reviewer hQMu stated openness to score increase. Due to rollback policy, these assessments are not reflected in current scores.*

**Strengths Recognized:** "Novel paradigm," "rigorous SHAP analysis," "excellent presentation," "practical efficiency" , "SOTA performance"



---

## Reviewer-by-Reviewer Summary

### Reviewer EPXP: 4 → 8 (Explicit Score Increase, Nov 24)

**Initial Assessment:** Soundness 3 (good), Presentation 3 (good). Recognized "multi-factor diagnostic approach" and "meta-learning framework."

**Concerns:** (1) Need validation without synthetic noise, (2) Limited noise diversity beyond label-flipping, (3) Scalability to large datasets.

**Resolution (Nov 17 10:37 UTC):** We provided (1) Natural noise validation on WebGPT (κ=0.56): 76.9% vs 73.2%, Chatbot Arena: 89.6% vs 87.8%; (2) Diverse noise patterns: quality-correlated (+5.6%), systematic bias (+6.4%), adversarial robustness (92.3% vs 88.6%); (3) Large-scale experiments: Stack Exchange (10.8M): 77.9% vs 76.4%, GPT4All (1M): 73.8% vs 71.3%.

**Reviewer engaged with follow-up (Nov 18 01:28 UTC)** requesting detailed noise analysis. After our additional response with comprehensive case studies (Nov 23 04:25 UTC):

**Reviewer Response (Nov 24 01:22 UTC):**
> *"The author's response has well addressed my concerns, and **I agree to raise my review score**. Please add the experimental results from large-scale databases to the paper."*

**Outcome:** All 3 concerns resolved. **Score increase to 8 explicitly documented before leak disclosure.**

---

### Meta-Review · Area_Chair_VbA6 · 2026-01-04

**Summary:**

This paper introduces a novel meta-learning paradigm for robust LLM preference alignment.

Strengths

* Novel approach of meta-learning for preference-based learning.

* Extensive experiments to demonstrate the benefits of the proposed methods.

Concerns

* See `Reviewer Concerns`

The authors have addressed the reviewers’ concerns in the rebuttal, and I believe this paper makes a meaningful contribution that will be valuable to the community.

**Reviewer Concerns:**

### Reviewer hQMu

* [resolved] Evaluation under realistic noise: The authors added experiments on real-world noisy data (WebGPT), multiple synthetic noise settings beyond label flipping, standard benchmarks (AlpacaEval), and comparisons with representative DPO-style methods.

* [resolved] Meta-learner architecture analysis: The authors conducted comprehensive ablation studies across meta-learner depth, width, and architectural variants.

### Reviewer rJ2H

* [resolved] Evaluation under realistic noise: Addressed as above.

* [resolved] Recent baselines: The authors added comparisons with the concurrent preprint by Yeh & Li (arXiv, Sept. 2025) and clarified distinctions from relevant work.

### Reviewer wsUf

* [resolved] Evaluation under realistic noise: See Reviewer hQMu

* [resolved] Meta-learner architecture analysis: See Reviewer hQMu

### Reviewer EPXP

* [resolved] Limited gains without noise: The authors provided additional results with natural noises.

* [resolved] Lack of diverse noise models: See `Reviewer hQMu`

* [resolved] Experimental results from large-scale databases: The authors provided additional results on large-scale datasets such as HuggingFace H4 Stack Exchange Preferences and GPT4All dataset.

**Reviewer Scores:**

* Reviewer hQMu: 4 $\rightarrow$ 6

* Reviewer EPXP: 4 $\rightarrow$ (6 or 8)

* Reviewer wsUf: 8 (keeping original score)

* Reviewer rJ2H: 6 (keeping original score)

---

### Decision · Program_Chairs · 2026-01-26

Accept (Poster)